# An assessment of ocean alkalinity enhancement using aqueous hydroxides: kinetics, efficiency, and precipitation thresholds

Mallory C. Ringham [1], Nathan Hirtle [1], Cody Shaw [1], Xi Lu [1], Julian Herndon [2,3], Brendan R. Carter [2,3], Matthew D. Eisaman [4,5]

[1] Stony Brook University, Stony Brook, NY, USA
[2] Cooperative Institute for Climate Ocean and Ecosystem Studies, University of Washington, Seattle, USA
[3] Pacific Marine Environmental Laboratory, National Oceanic and Atmospheric Administration, Seattle, WA, USA*
[4] Department of Earth & Planetary Sciences, Yale University, New Haven, CT, USA
[5] Yale Center for Natural Carbon Capture, Yale University, New Haven, CT, USA

*Correspondence to*: Mallory Ringham (mallory.ringham@stonybrook.edu); Current address: Ebb Carbon Inc., San Carlos, CA, USA

## Abstract

Ocean alkalinity enhancement (OAE) is a promising approach to marine carbon dioxide removal (mCDR) that leverages the large surface area and carbon storage capacity of the oceans to sequester atmospheric $CO_2$ as dissolved bicarbonate ($HCO_3^-$). One OAE method involves the conversion of salt in seawater into aqueous alkalinity (NaOH), which is returned to the ocean. The resulting increase in seawater pH and alkalinity causes a shift in dissolved inorganic carbon (DIC) speciation toward carbonate and a decrease in the surface-ocean $pCO_2$. The shift in the $pCO_2$ results in enhanced uptake of atmospheric $CO_2$ uptake by the seawater due to gas exchange. In this study, we systematically test the efficiency of $CO_2$ uptake in seawater treated with NaOH at aquaria (15L) and tank (6000L) scales to establish operational boundaries for safety and efficiency in advance of scaling up to field experiments. $CO_2$ equilibration occurred on order of weeks to months, depending on circulation, air forcing, and air bubbling conditions within the test tanks. An increase of ~0.7-0.9 mol DIC/ mol added alkalinity (in the form of NaOH) was observed through analysis of seawater bottle samples and pH sensor data, consistent with the value expected given the values of the carbonate system equilibrium calculations for the range of salinities and temperatures tested. Mineral precipitation occurred when the bulk seawater pH exceeded 10.0 and $\Omega_{aragonite}$ exceeded 30.0. This precipitation was dominated by $Mg(OH)_2$ over hours to 1 day before shifting to $CaCO_{3,\ aragonite}$ precipitation. These data, combined with models of the dilution and advection of alkaline plumes, will allow for estimation of the amount of carbon dioxide removal expected from OAE pilot studies. Future experiments should better approximate field conditions including sediment interactions, biological activity, ocean circulation, air-sea gas exchange rates, and mixing-zone dynamics.

## Keywords

Ocean Alkalinity Enhancement (OAE); marine carbon dioxide removal (mCDR); ocean carbon dioxide removal (ocean CDR)

## 1 Introduction

The Sixth Assessment Report of the Intergovernmental Panel on Climate Change reported that in addition to a drastic decrease in $CO_2$ emissions, active removal of 5-15 Gt of atmospheric $CO_2$ per year by 2100 is necessary to constrain average global warming to less than 1.5 - 2 °C (noting that the magnitude of carbon removals varies by

climate scenario: IPCC, 2022; Rogelj, 2018). A wide variety of negative emissions technologies (NETs) are under
development to meet this enormous challenge (Minx et al., 2018; NASEM, 2019; NASEM, 2021; Rueda et al.,
2021; Vitillo et al., 2022).
A suite of promising approaches to $CO_2$ removal termed ocean or marine carbon dioxide removal (ocean CDR or
mCDR, respectively) leverage the enormous surface area and carbon storage capacity of the ocean (Boettcher et al.,
2019; NASEM, 2021). Ocean alkalinity enhancement (OAE) is an mCDR method that aims to store atmospheric
$CO_2$ in a dissolved phase in the ocean as bicarbonate ions ($HCO_3^-$), thereby accelerating a natural planetary $CO_2$
regulation mechanism, the carbonate-silicate cycle (Berner, 1983; Isson et al., 2020). OAE has the potential to scale
to gigatons of $CO_2$ removal per year (He and Tyka, 2023), but development of this approach requires careful
consideration of: the methods and materials used to source and process alkalinity; the form and method of delivery
of alkalinity to the surface ocean (for example, aqueous or solid phase); and selection of appropriate geographic sites
for alkalinity dispersal (Oschlies et al., 2023). OAE methods under exploration include: mining and crushing
alkaline minerals (e.g., olivine, basalts) to be spread via ship or in coastal environments (e.g., beach restoration, or
salt marsh distribution) (Feng et al., 2017; Köhler, Hartmann, and Wolf-Gladrow, 2010; Monserrat et al., 2018;
Rigopoulos et al., 2018); the mining or industrial production of $Mg(OH)_2$ or mining $CaCO_3$ and calcining it to CaO
or $Ca(OH)_2$, with the $Mg(OH)_2$ or $Ca(OH)_2$ spread via ship or coastal outfall pipe (Harvey, 2008; Ilyina et al., 2013;
Kheshgi, 1995; La Plante, 2023; Moras et al., 2022; Nduagu, 2012; Rau, 2008; Renforth and Henderson, 2017;
Shaw, 2022); and the electrochemical conversion of saltwater into aqueous hydroxides and dispersal via coastal
outfalls (de Lannoy et al., 2018; Eisaman et al., 2018; Lu et al., 2022; Tyka, Van Arsdale, and Platt, 2022; Eisaman
et al., 2023).
Many of these approaches and technologies are at a nascent stage. We must move quickly to quantitatively test and
characterize their performance to determine which, if any, justify larger-scale deployment. The electrochemical
conversion of salt (NaCl) into aqueous alkalinity (NaOH) has many potential advantages in scaling considerations,
including simplified distribution of a liquid product to the ocean, avoidance of mining and the transportation of the
alkalinity source over long distances, and avoidance of potentially harmful impurities present in mined alkalinity
sources (NASEM, 2021; Caserini, Storni, and Grosso, 2022).
Total alkalinity (TA) is defined as the excess of proton acceptors over proton donors in an aqueous solution (Eq. 1),
where ellipses represent neglected acids and bases (Dickson 1981; Dickson 1992; Wolf-Gladrow et al., 2007). A
higher TA value for a seawater sample indicates that it has a higher buffering capacity than a sample with a lower
TA value. That is, for sample with a higher TA value, the addition of a given amount of acid to the sample will
decrease its pH less than for a sample with a lower TA value.
$$TA = [HCO_3^-] + 2\,[CO_3^{2-}] + [B(OH)_4^-] + [OH^-] + [HPO_4^{2-}] + 2\,[PO_4^{3-}] + \ldots - [H^+] - [HSO_4^-] - \ldots \quad (1)$$
From Eq. (1), we see the increased $OH^-$ concentration in a treated seawater solution corresponds to a salt solution
with increased alkalinity relative to the starting salt solution. This increase in $OH^-$ ion concentration rapidly
increases the seawater pH upon mixing, resulting in a shift of the dissolved inorganic carbon (DIC) speciation
towards carbonate (Eisaman et al., 2023):
$$OH^- + HCO_3^- \rightarrow H_2O + CO_3^{2-} \qquad\qquad (2a)$$
$$OH^- + CO_{2,aq} \rightarrow HCO_3^- \qquad\qquad (2b)$$
The concentration of dissolved $CO_2$ gas ($CO_{2,aq}$) in this alkalinity-enhanced seawater is less than it would be if it
were in equilibrium with atmospheric $CO_2$ (Equation 2b). Over the longer timescale required for air-sea gas
exchange - weeks to months (Wang et al., 2023) or months to years (He and Tyka, 2023) depending on location - the
disequilibrium in the surface ocean resulting from the alkalinity addition drives the invasion of atmospheric $CO_2$
into seawater (or lessens the outgassing of $CO_2$ from the surface ocean to the atmosphere), where it reacts with
carbonate and is stored primarily in the stable bicarbonate phase (Jones et al., 2014; Bach et al., 2023; Renforth and
Henderson, 2017; Eisaman et al., 2023).
$$CO_{2,g} \rightarrow CO_{2,aq} \tag{3a}$$
$$CO_{2,aq} + CO_3^{2-} + H_2O \rightarrow 2HCO_3^- \tag{3b}$$
Under typical ocean conditions, after equilibrium has been reached, OAE results in an increase in the DIC in
seawater on the order of 0.7-0.9 moles of DIC per mole of NaOH added, with a slightly increased pH relative to the
initial value (He and Tyka, 2023).
It is possible that air-sea gas exchange will not completely drive the seawater $pCO_2$ to the initial unperturbed value
before the seawater sinks into the ocean interior and loses contact with the atmosphere for hundreds to thousands of
years. Therefore, the DIC anomaly relative to the alkalinity anomaly present when the seawater sinks into the ocean
interior may be used to assess the effective impact of the OAE for capturing atmospheric $CO_2$ on the 0-100 year
timescales that are most important for climate interventions.
In addition to the storage of atmospheric $CO_2$ in the form of DIC, this process may have the potential to locally and
transiently mitigate the elevated $pCO_2$ associated with ocean acidification (NASEM, 2021; Cross et al., 2023;
Butenschön et al., 2021). In a water body with a finite seawater exchange rate with the ocean, such as a semi-
protected estuary or bay, alkalinity could be added in a controlled manner such that the combination of the rapid
reactions described by Eq.(1) and the exchange/flushing rate with the open ocean result in the bay being held in
steady-state at a target pH or aragonite saturation state value that is higher than its equilibrium value under
conditions of ocean acidification. As this added alkalinity diffuses through the bay and makes its way to the open
ocean, $CO_2$ removal and storage as DIC would occur. By metering the rate of alkalinity addition to the bay to match
the flushing rate, the pH or saturation state of the bay can be held at a constant target value. Even once equilibrium
has been achieved in the open ocean, the pH and the carbonate ion concentration in the open ocean remains slightly
higher than before the alkaline discharge. However, the absolute value of this pH increase after equilibrium has been
reached is sufficiently small relative to the alkalinity and DIC increase that mitigating ocean acidification on a
global scale with this method is unfeasible. For example, increasing the equilibrium pH value from 8.0 to 8.1 at a
fixed $pCO_2$ of 400 µatm (at 20 C and 35 salinity with no macronutrients) requires a TA increase of around ~620
µmol/kg-sw and a DIC increase of around 520 µmol/kg-sw. Using these numbers, mitigating OA over the entire 360
million $km^2$ surface of the ocean to a depth of 100 meters would require around 487 gigatons of cumulative $CO_2$
removal. Deploying SEA MATE in the ocean or coastal waters will require an understanding of carbonate chemistry
in seawater in the ocean volume under consideration, as well as thresholds for safe operation. For example, at the
point of alkaline dispersal where there is the maximum change in seawater chemistry, SEA MATE must control the
rate of alkalinity addition relative to the rate of mixing and dilution in the ocean to avoid the precipitation of
$Mg(OH)_2$ or $CaCO_3$ (Hartmann et al., 2023; Moras et al., 2022). While $Mg(OH)_2$ readily redissolves, an increase in
turbidity due to precipitation may negatively affect marine organisms (Bainbridge et al., 2018; Broderson et al.,
2017). By contrast, $CaCO_3$ will generally not redissolve in the surface ocean without biological mediation, and
runaway precipitation, where alkalinity removed by precipitation exceeds that added by the OAE treatment, can
occur under conditions of increased aragonite saturation state and increased nucleation sites in the water column
(Moras et al., 2022). $CaCO_3$ precipitation could counteract the intended effect of the OAE intervention by removing
alkalinity from the surface ocean and releasing $CO_2$ gas via Eq. 5 (Zeebe and Wolf-Gladrow, 2001):
$$Ca^+ + 2HCO_3^- \rightarrow CaCO_3(s) + CO_2(g) + H_2O \tag{5}$$
Upon dispersal to the ocean through a coastal outfall pipe, the added alkalinity is advected and diffuses away from
the point source, becoming increasingly diluted through the mixing zone. Because the timescale for air-sea gas
exchange and re-equilibration described by Eq. (2) is longer than the characteristic timescale for dilution driven by
tides, currents, and weather, most of the $CO_2$ removal occurs far from the mixing zone. Dilution will spread the
impacts over a broad area, to an extent that it is unlikely that the impacts on the DIC distribution can be quantified
using only direct measurements, given current instrument resolution and the typical dynamic range of natural
variability (Wang et al., 2023). In general, options for measurement, reporting, and verification (MRV) of OAE will
therefore rely on (Ho et al., 2023): experimentation in laboratory and mesocosm settings, such as the work we
describe here, to establish $CO_2$ removal dynamics under conditions of OAE; direct monitoring of the rate and
characteristics of alkalinity addition into seawater; monitoring the seawater carbonate and environmental chemistry
in the immediate vicinity of the outfall via sensors and sampling (Cyronak et al., 2023; Schulz et al., 2023); and
ocean modeling to estimate CDR beyond the range of direct detection (Fennel et al., 2023).
While some work has investigated various aspects of NaOH-based ocean alkalinity enhancement in microcosms
(Ferderer et al., 2022; Hartmann et al., 2023), and mesocosms (Groen et al., 2023), and other work has studied the
release of NaOH over natural coral reefs as a method of local ocean acidification mitigation (Albright et al., 2016), a
systematic characterization of the efficiency and kinetics of OAE as a function of key process parameters has not yet
been performed. Here we report the first tank-scale tests of OAE that use aqueous hydroxide (NaOH) to enhance the
alkalinity of natural seawater, a process that mimics OAE via the electrochemical brine-to-alkalinity conversion
used in the SEA MATE process. Our experiments, conducted in 6,000 liter tanks using seawater pumped from Flax
Pond on Long Island Sound in Stony Brook NY, quantify the magnitude and timescale of the $CO_2$ removal from the
air and storage as seawater DIC by monitoring the air-seawater re-equilibration after an initial alkalinity
perturbation. In addition, our use of both laboratory-processed bottle samples and field-deployable sensors to
measure and over-constrain the carbonate chemistry response allows us to assess the suitability of certain sensing
platforms for MRV. Finally, we investigate safe thresholds for the rate and concentration of alkalinity addition to
avoid: (1) the precipitation and redissolution of $Mg(OH)_2$ that can lead to local, temporary increases in turbidity; and
(2) the precipitation of $CaCO_3$, which partially reverses the intended OAE effect by removing alkalinity from, and
releasing $CO_2$ gas into, the surrounding seawater.
Using this approach, we address the following key questions:
(1) How much additional atmospheric $CO_2$ is stored in seawater as DIC in response to a given alkalinity
perturbation?
(2) What is the timescale for $CO_2$ removal from the air, and how does it depend on pH and the magnitude of
alkalinity enhancement?
(3) What are the conditions for $Mg(OH)_2$ precipitation upon addition of NaOH to seawater?
(4) What are the threshold values for pH and aragonite saturation state beyond which undesired $CaCO_3$ precipitation
will occur?
Answering these questions is key to assessing the viability of this approach and to optimizing its eventual
deployment. The large tank experiments presented in this manuscript provide a stepping stone between bench-scale
experiments and in-situ mesocosms or field pilots. Even if these experiments simply confirm stoichiometric and
modeled expectations, this is critical information in the design and implementation of OAE deployments. This work
is a necessary part of the growing scientific body that will allow for field trials to progress.
**2. Methods**
**2.1 Experimental procedure**
We investigated the carbonate chemistry changes resulting from the addition of $NaOH_{(aq)}$ to natural seawater over
timescales ranging from 2 weeks to 2 months in a series of experiments at two scales: (1) two large (~6200 L)
indoor tanks, and (2) multiple 15 L aquaria (Fig. 1). The large volume of tank experiments allowed for precise
measurement of the seawater carbonate chemistry via bottle sampling (1L each, sent to NOAA/PMEL for analysis)
with high sampling frequency. To compliment these measurements, we also performed a series of experiments in
smaller aquaria (15 L each), which enabled a larger number of replicates and a faster time to equilibrium when
bubbled with air.

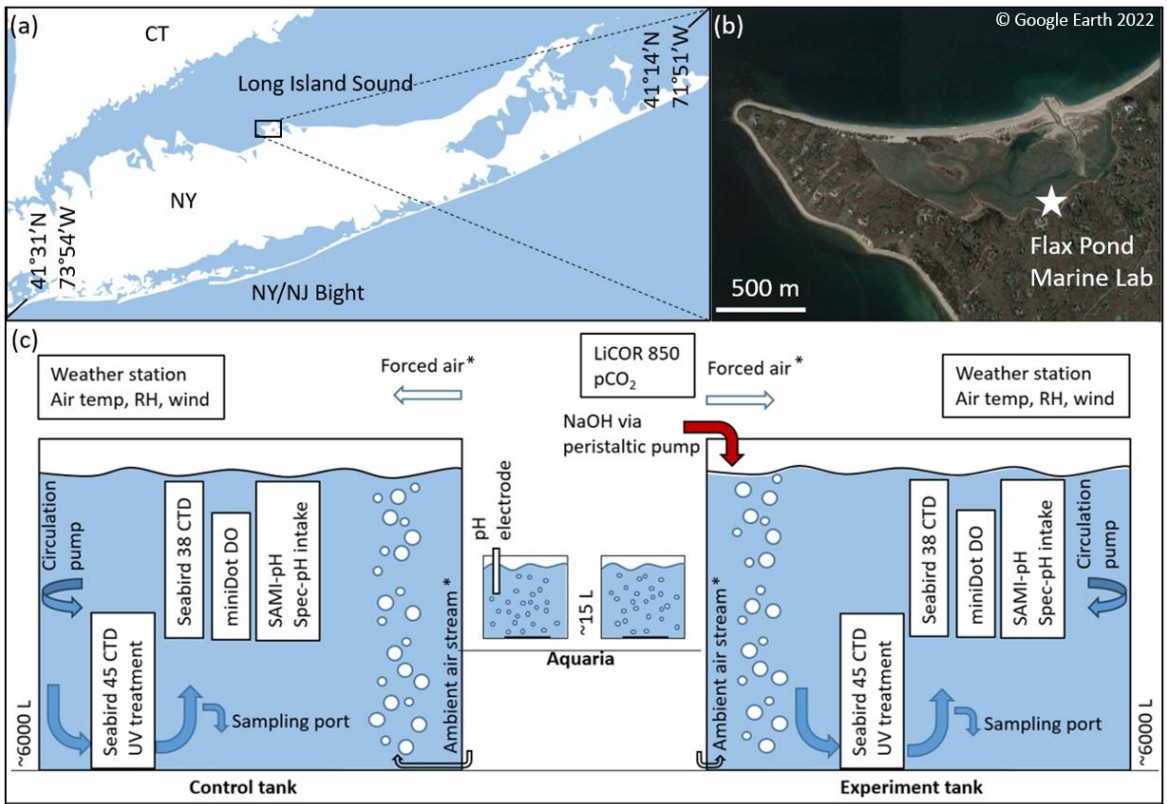


**Figure 1:** (*a, b*) Flax Pond Marine Laboratory is located on Long Island Sound, New York, USA (© Google Earth
2022). (*c*) The ~6000 L control and experiment tanks were instrumented with a series of oceanographic sensors and
sampled routinely for DIC/ TA analyses to allow for measurement of carbon uptake over time following an addition
of alkalinity in the form of NaOH. The ~15 L aquaria were instrumented with standard pH electrodes and monitored
with routine TA analyses. The Forced air* and Ambient air streams* indicate their use in some but not all
experiments, as noted in later sections.

This study was conducted at the Flax Pond Marine Laboratory at Stony Brook University, NY. All experiments used
natural seawater collected from Flax Pond, part of a 128-acre salt marsh tidal wetlands connected to the Long Island
Sound. The surface areas of the tanks and aquaria were ~4.6 $m^2$ and ~0.1 $m^2$, respectively. The tanks had a diameter
of 2.4 m, a total height of 1.52 m, and were typically filled to a height of ~1.35 m, allowing for a corresponding
seawater volume of 6185 L. The aquaria had a diameter of 0.3 m and were typically filled to a height of ~0.23 m, for
a total seawater volume of 15 L. The large tank volumes were chosen to limit interactions with walls while
increasing the air-seawater boundary, and to lose a smaller fraction of their volume to evaporation. These tanks
allow for in-situ oceanographic sensor deployment and frequent bottle sampling while retaining semi-controlled
temperature, mixing, filtration, and biological control. The inherent limitations of these tank tests include limited air-
sea interaction, unrealistic light levels and circulation, and biological responses that are not a perfect representation
of natural seawater in the ocean, but serve as a stepping stone to mesocosm and eventual field experiments. On
average, the large (~6,000 L) tank experiments took ~6.5 weeks after dosing with NaOH to reach 90% of the
calculated or extrapolated asymptotic ΔDIC/TA addition ratio indicative of full air-seawater equilibrium, as will be
discussed in Section 3. Therefore, in addition to the large tank tests, we conducted a series of smaller aquaria
alkalinity additions to increase our capacity for experimental test cases. The limitations of the aquaria include
limited sensor options, unrealistic circulation, and limited biological control. While it is expected that equilibration
occurs more rapidly in the small aquaria than in the large tanks, the results from these cases should be similar as
$CO_2$ equilibrates across the air-sea boundary. However, we note that some variation is expected due to limited
sensing and sampling options in the smaller aquaria and the greater potential for biological growth in the large tanks
over longer timescales.

### 201    2.1.1 Tank experiments

Seawater was pumped into the tanks at high tide through a series of sock filters to exclude macroscopic biology. The
tanks were then dosed to 40 ppm bleach (sodium hypochlorite) and the shock-treated seawater was allowed to
circulate through the tanks for ~1 day to limit biological growth. The seawater was then circulated through UV light
arrays to break down the bleach over ~1-2 weeks, as assessed by a standard Hach test kit for free chlorine. During
this period, seawater was pumped between the two large test tanks (~25 L/min) to increase mixing of the bleach and
to homogenize the tanks to similar initial conditions. For the remainder of each experiment, the seawater was
continually pumped through the UV sterilizers. Measurements of total alkalinity showed no significant differences
in the bulk seawater TA before and after the bleaching process in any experiment or control tank. In an early
experiment (in which bulk $pH_T$ was raised from the initial condition to 8.7, as described below), the initial $pH_T$ and
DIC varied between the control and experiment tanks by 0.17 and 77 µmol kg$^{-1}$, respectively. This was because
seawater was pumped from Flax Pond into multiple reservoirs and was then unevenly distributed between the tanks.
The experiments were subsequently refined to allow for several days of cross-pumping between tanks to
homogenize the control and experiment seawater before NaOH was added at the start of an experiment. More details
on experimental variations and a larger summary table are available in the Supplementary Materials.
Oceanographic sensors and discrete daily bottle sampling, as described in Sections 2.2 and 2.3, respectively, were
deployed for carbonate chemistry analysis for several days prior to the alkalinity addition to understand the initial
baseline conditions in both tanks. Two submerged pumps were used for water circulation within each tank: the first
pump (Current eFlux DC Flow Pump, 210 GPH) cycled seawater through the UV arrays with an estimated
overturning time of the bulk tank on order of 1 day, and a second (Kedsum Submersible pump, 260 GPH), mounted
at an angle halfway down the tank wall, allowed for subsurface circulation within the tank to reduce the occurrence
of unmixed 'dead zones ' and subsequent non-homogenous biological growth, as assessed visually on the surface of
the water and/or tank lining. Initial tank experiments were conducted with a still surface condition, i.e., with no
visible water movement across the surface of each tank. As experiments progressed, forced air movement was added
across the surface of each tank using a stationary fan with a wind speed of ~5 kph. This was done to control for
potential variations in the laboratory HVAC system and to potentially reduce the time to equilibration for the
experiments by increasing the rate of air-sea $CO_2$ equilibration. In later experiments, air was bubbled into the bottom
of each tank at a rate of ~ 30 L min$^{-1}$ with an estimated surface area of ~ 0.3 m$^2$, with a goal of further increasing the
rate of equilibration to allow for more rapid throughput of experiments. These variations are further discussed in
Section 2.4.
After baselining, one tank (referred to as the "experimental tank") was dosed with enough 0.5 M NaOH (see
Supplementary Materials) to raise the bulk seawater pH to the target pH of interest for a given experiment, and the
same volume of DI water was added to the other tank (referred to as the "control tank"). NaOH additions were
typically dosed into the tank via peristaltic pump at a low enough rate (~50 mL/min) that a steady increase in bulk
tank pH was observed, but local pH measured just below the NaOH introduction never exceeded a pH of 9.0. A
pump (~25 L/min) was placed just below the NaOH stream to speed the mixing of NaOH into the bulk tank,
increase dilution from the point source, and to prevent the immediate precipitation of $Mg(OH)_2$ upon contact of the
NaOH with seawater. This pump was removed after the full volume of NaOH was mixed into the tank.
After the alkalinity addition, the tanks were left to equilibrate with the atmosphere and were monitored by sensors
and sampling as described in Sections 2.2 and 2.3. The tanks were indoors in the wet laboratory at Flax Pond Marine
Lab, such that temperature and $CO_2$ concentration were moderated by the building's HVAC system, but varied
throughout days and seasons depending on other uses of the lab space. The experiments were concluded when the
observed pH or DIC (calculated from daily pH and frequent TA measurements) appeared to stabilize (e.g., ΔpH
±0.05% or ΔDIC ±10 μmol $kg^{-1}$ per day) over several days. The continuous improvement of experimental methods
during this study resulted in some minor variations among the methods used for each experiment, including methods
of NaOH dosing, tank circulation, and biological control, as discussed where necessary in Section 3 and in the
Supplementary Materials.
**2.1.2 Aquaria experiments**
A series of polycarbonate aquaria were filled with 15 L of seawater taken from the large control tank just after the
described bleaching and bleach breakdown procedure was completed. NaOH was dosed into each aquarium to reach
a targeted bulk $pH_T$, with a corresponding volume of DI $H_2O$ added to the control aquarium, and then the seawater
was allowed to equilibrate with atmospheric $pCO_2$ over days to weeks. The aquaria did not have either UV light
arrays for biological control or aquarium pumps for internal circulation. With the exception of a single target $pH_T$ 8.5
experiment, all aquaria were bubbled with ambient air (~4 L $min^{-1}$) via a standard aquarium bubbling bar spanning
the center diameter of each aquarium, allowing for rapid $CO_2$ exchange . There was no fine control on air bubbling,
but the surface area of all air bubbles in a given aquarium at any point in time was estimated at ~0.01 $m^2$. No sensors
were deployed in the aquaria due to their limited size, and seawater chemistry was established via discrete $pH_T$ and
TA measurements (Sect. 2.2). An optically clear lid was placed on each aquarium to reduce evaporation and
splashing onto nearby equipment. Some evaporation was evident from the rising TA throughout these experiments,
but was not resolvable within the resolution of a handheld salinometer used for these experiments. Temperature was
discretely recorded from a combination Ross pH electrode.
As shown in Eq. (6), we define the dimensionless 'Carbon-to-Alkalinity Ratio' (CAR) for our experiments as the
molar ratio of the increase in $n$DIC (in units of μmol/kg, normalized to the system's initial salinity to account for
evaporation) to the magnitude of the TA increase (ΔTA, in units of μmol/kg). $n$DIC$_{equ}$ is the measured (via direct
titration) or calculated (via CO2SYS using measured TA and $pH_T$) DIC value that the system reached at the end of
an experiment (Pierrot et al., 2006; Van Heuven et al., 2011). Some experiments were left long enough to achieve
equilibration with atmospheric $CO_2$, but others were halted early. In these cases, a CO2SYS calculation was used to
estimate the DIC increase expected at equilibration given initial seawater conditions, and the difference between this
value and the final recorded $n$DIC$_{equ}$ was used to estimate the overall percent equilibration for a given experiment.
Depending on experimental constraints described in later sections, $n$DIC$_i$ may represent either: (1) the final $n$DIC
measured (via titration of bottle samples) or calculated (via CO2SYS using seawater TA and pH) in the control tank,
or (2) the 'baseline' $n$DIC before the addition of NaOH to a given aquaria experiment, for cases where a
corresponding control case may not be available.  Note that because we are reporting CAR values where the
measured DIC has reached or has been estimated at equilibrium, the CAR values we measure and report reflect the
ratio of ΔDIC to ΔTA that would be expected given sufficient time for air-sea exchange to reach equilibrium, and so
are equivalent to directly measuring the value of the "TA addition potential impact ratio" as defined by Wang et al.,
277    2023.

278       Carbon-to-Alkalinity Ratio (CAR) = ($n$DIC$_{equ}$ - $n$DIC$_i$) / ΔTA                    (6)

## 2.2 Oceanographic sensors

Each tank was instrumented with a series of sensors placed halfway down the wall of the tank near the inlet of the UV circulation pump. A Seabird 38 Digital Oceanographic Thermometer and Seabird 45 MicroTSG Thermosalinograph continuously monitored seawater temperature and salinity, respectively. Dissolved oxygen was measured by a PME miniDOT Logger at 10 min resolution. $pH_T$ was monitored daily by a SAMI-pH (manufacturer specified accuracy/precision ~ 0.003/0.001, though this accuracy is likely an underestimate of the uncertainty given known challenges for the calibration of the $pH_T$ measurements) and by a semi-automated spectrophotometric (spec-pH) pH unit (~± 0.0055/0.0004) as described by Carter et al. (2013). CRM measurements were taken by each pH system at the beginning and end of each experiment and were used alongside discrete samples of DIC and TA as described in Section 2.3 to constrain the stability of each sensor. The SAMI-pH measurements were recorded at ambient seawater temperature and corrected for in-situ salinity as recorded by the Seabird Thermosalinograph following best practices from the manufacturer. The spec-pH analyses occurred in a jacketed cuvette held at 20 °C (regulated via water bath) and were corrected to the in-situ bulk tank temperature and salinity as recorded by the Seabird Thermometer and Thermosalinograph. Both the SAMI-pH and spec-pH rely on spectrophotometric analysis of metacresol purple indicator dye, which allows for pH measurement within the $pH_T$ range of approximately 7 to 9. For experiments in which enough NaOH was dosed into seawater to raise pH above these limits, a Thermo Scientific Orion ROSS Ultra pH/ATC Triode combination electrode (8157BNUMD) was used to monitor $pH_{NBS}$ at the surface of the tank (±0.01 precision), which was then converted to $pH_T$ for comparison with the other pH sensor systems.

A LiCOR LI-850 sensor was used to analyze atmospheric $pCO_2$ (±1.5% accuracy) above the tanks. The inlet to this sensor was periodically moved between tanks to ensure that atmospheric $pCO_2$ in the vicinity of the control and experiment tanks was the same. AcuRite Iris weather stations were mounted on the side of each tank to monitor air temperature (±2 °C), relative humidity (±3%), and air speed (±0.8 m s$^{-1}$). All data were compiled on an hourly basis in a custom R package.

## 2.3 Discrete sampling

Two types of discrete sampling were used to constrain carbonate chemistry throughout these experiments. First, 500 mL of seawater was collected and preserved from each tank, typically on a daily basis, and as frequently as hourly during the addition of NaOH, following best practices laid out by Dickson (2007) including overflowing of the sample bottles during collection and addition of 0.2 mL of saturated mercuric chloride ($HgCl_2$) as a preservative. These bottle samples were analyzed for DIC and TA at NOAA Pacific Marine Environmental Laboratory (NOAA/PMEL). DIC concentrations were measured using a coulometer (UIC Inc.) and Single Operator Multiparameter Metabolic Analyzer (SOMMA) (Johnson et al., 1985). TA was determined by an open-cell acidimetric titration (Dickson et al. SOP 3b, 2007). The accuracy of DIC and TA measurements was assessed with Certified Reference Materials (CRMs, supplied by the Dickson laboratory at Scripps Institution of Oceanography), and overall uncertainty for both DIC and TA was typically ± 0.1% (~2 μmol/kg).

In addition, discrete seawater samples were analyzed for TA via open-cell potentiometric titration at Stony Brook University. A Thermo Scientific Orion ROSS Ultra pH/ATC Triode combination electrode (8157BNUMD), calibrated using three buffer solutions ($pH_{NBS}$ 4.01, 7, and 10.01) was used to track the titration of a ~20 mL seawater sample with a dilute HCl solution (~0.1 M in 0.7 M NaCl, calibrated daily with CRM or a secondary seawater standard) following a modified Gran titration procedure using a Kloehn digital syringe pump (Song et al., 2020; Wang and Cai, 2004). The precision of TA measurements was ~±5-10 μmol/kg. This TA data was corrected to that of the bottle samples analyzed via titration at NOAA PMEL where available (see Supplementary Materials).

There are several differences between the aquaria experiments and the larger tank experiments. First, the aquaria experiments were monitored daily to every few days by discrete measurement of TA at Stony Brook University and

$pH_{NBS}$ via Thermo Scientific Orion ROSS Ultra pH/ATC Triode combination electrode (8157BNUMD) (±0.01
precision), which was then converted to $pH_T$ and corrected against the other pH sensor systems via occasional bottle
samples for DIC and TA analysis at NOAA PMEL. Variations between these experiments are noted in Section 3
where necessary and in the Supplementary Materials.
In either tank or aquaria cases where mineral precipitation was observed, 0.5 – 1 L of seawater was vacuum filtered
through a 0.45 μm Whatman GF/F filter via vacuum pump and the solids were rinsed with DI water 3 times to
remove NaCl. The precipitate was dried in an oven at 90 °C, then crushed into a uniform powder via mortar and
pestle. Samples were analyzed via Hitachi 4800 Scanning Electron Microscopy (SEM) (5 kV) and Rigaku SmartLab
X-ray Diffraction (XRD) (Cu Kα, 1.5406 Å, 10 - 100° 2 θ at 4°/min) at Brookhaven National Laboratory at the
Materials Synthesis and Characterization Facility of the Center for Functional Nanomaterials.
**2.4 Evaluation of $CO_2$ uptake by seawater in response to NaOH perturbation**
Seawater carbonate chemistry measurements were used to analyze the uptake of $CO_2$ in each tank, primarily relying
on calculations from the NOAA/PMEL DIC and TA analyses of bottle samples when available and using sensor pH
and Stony Brook TA measurements for cross-verification or to fill in between discrete DIC samples. DIC and TA
data were normalized to the salinity at the start of a given experiment to account for evaporation (Friis et al., 2003).
Carbonate chemistry calculations were then performed using CO2SYS (Lewis and Wallace, 1998), with Lueker et
al. (2000) carbonate constants, Dickson (1990) for $KSO_4$, and Lee et al. (2010) for total boron. Wherever possible, a
combination of CRM analyses and comparisons between simultaneous pH sensor and NOAA PMEL bottle samples
were used to correct SAMI-pH and spectrophotometric pH sensor data for drift.
Changes in the seawater carbonate chemistry over time were analyzed with respect to shifts away from the baseline
within a single control or experiment tank, as well as with respect to the differences between the control and
experimental tanks.
**3 Results and Discussion**
**3.1 Large tank experiments**
A summary of the range of oceanographic variables measured by sensors and bottle samples, calculated via
CO2SYS, or extrapolated to equilibration conditions during the large tank experiments is provided in Table 1. This
summary includes 6 experiments including 3 targeting $pH_T$ 8.5 (still surface water, with forced air, and with forced
air and air bubbling) and one (each) targeting $pH_T$ values of 8.7 (still surface water), 9.5 (with forced air and air
bubbling), and 10.3 (still surface water). While the initial seawater conditions were similar between the control and
experiment tanks, we note that these cases are not entirely comparable after the termination of cross-pumping
between tanks and the subsequent addition of alkalinity. While tanks were initially bleached, eventually some
biological growth was noted in each tank with potential differences in spatial and temporal distribution as well as
species and community differences. Herein, we assume that differences between the control and experiment cases
are due to the addition of alkalinity alone, but we note that characterization of other potential confounding factors is
a subject for future work.
The initial $pH_T$, TA, and DIC varied across experiments as seawater was collected between March 2022 and May
2023, ranging from pH 7.66 (December 2022) – 7.95 (May 2023), TA 2001 (May 2023) – 2176 (March 2023)
μmol/kg, and DIC 1847 (May 2023) – 2021 (March 2023) μmol/kg. Both measured and CO2SYS -calculated DIC
and TA values were normalized to salinity to account for evaporation, which drove salinity increases ranging from
0.2 – 7.1 across these experiments.
After the addition of NaOH, the control and experiment tanks were allowed to equilibrate with atmospheric $CO_2$.
While refinements in the experimental design allowed for complete or near-complete equilibration in later
experiments, as determined by the stabilization of nDIC at some asymptotic value, early experiments were
terminated before full equilibration. In all experiments, the absorption of atmospheric $CO_2$ began immediately after
the NaOH addition, as determined by decreasing pH and $\Omega_{arag}$ and increasing DIC and seawater $pCO_2$. nTA was
fairly stable or increasing (+10 - 60 μmol kg$^{-1}$) after the NaOH addition in all cases except the $pH_T$ = 10.3
experiment, where nTA and DIC rapidly decreased due to runaway $CaCO_3$ precipitation. A stable TA value is an
indicator that no significant persistent mineral precipitation (e.g., $Mg(OH)_2$ or $CaCO_3$) has occurred. In the absence
of active mixing or bubbling, $Mg(OH)_2$ precipitation occurred immediately upon the introduction of NaOH to
seawater, however the precipitation can be rapidly dissolved by turbulence (i.e., pumping NaOH directly above a
strong circulation pump and/or stream of air bubbles). No $CaCO_3$ precipitation was observed in the tanks or aquaria
for which the bulk seawater $pH_T$ was <10.0. The $pH_T$ = 10.3 experiment was designed to induce $CaCO_3$ runaway
precipitation, as described in Section 3.3.
$\Omega_{arag}$ ranged from 1.4 - 2.5 in the control tanks with minimal variation over the course of any given experiment.
During the three experiments in which bulk $pH_T$ was increased to ~8.5, $\Omega_{arag}$ increased immediately to 6.0 - 6.3 at the
peak of the experiments, before slowly decreasing to 2.8 - 3.0 as the seawater equilibrated with atmospheric $CO_2$.
For the bulk $pH_T$ 9.5 experiment, $\Omega_{arag}$ increased to 20.2 and slowly decreased to 5.0 when the experiment was
ended at full equilibration. Mineral precipitation was observed in the bulk $pH_T$ 10.3 experiment, where $\Omega_{arag}$ was
increased to 30.3 and rapidly (<1 week) fell to 5.2 after the addition of NaOH.
The results of one representative set of time-series measurements from the control and experiment tanks are shown
in Figure 2 for the case where $pH_T$ of the bulk experiment tank was raised to 8.5 then allowed to relax into
equilibration with the atmosphere without the addition of surface air forcing or bubbling. Time-series plots for the
other tank-scale experiments are available in the Supplementary Materials.

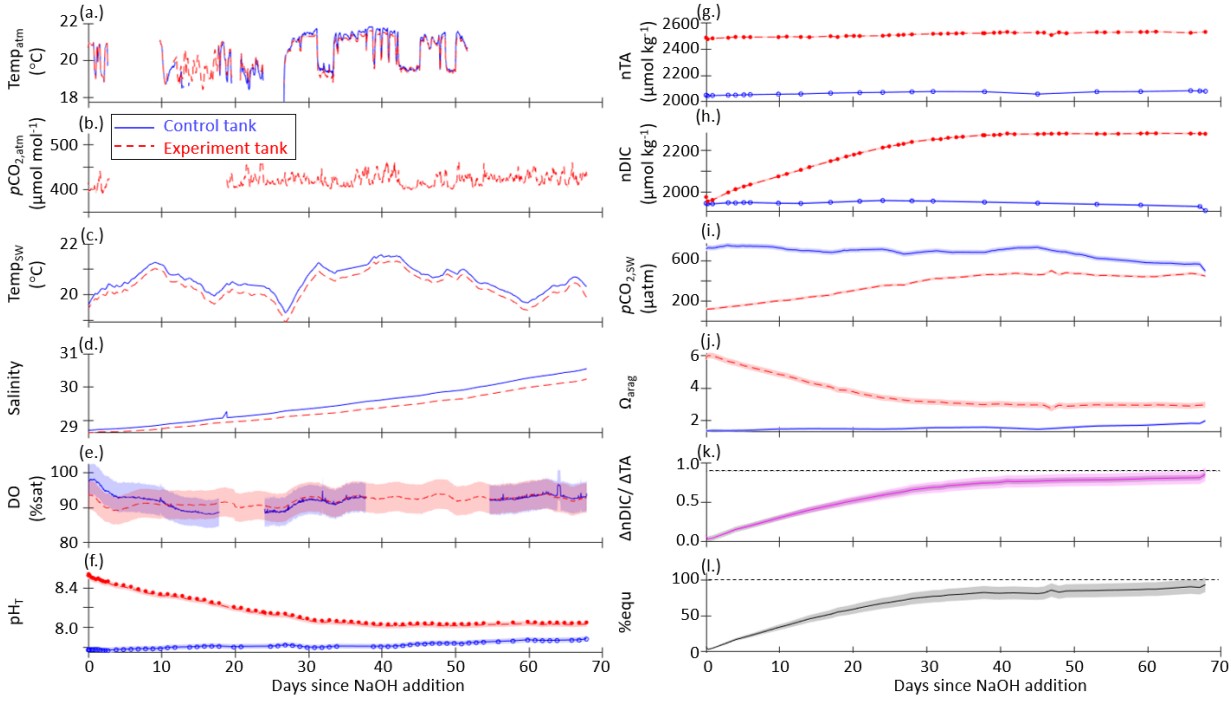


**Figure 2:** Time-series data for the case where $pH_T$ of the bulk experiment tank was raised to 8.5 with no forced air flow and no bubbling (still surface) for control (blue, solid) and experiment (red, dashed) tanks: (*a*) continuously measured air temperature, (*b*) atmospheric $pCO_2$, (*c*) seawater temperature, (*d*) salinity, and (*e*) dissolved oxygen; (*f*) $pH_T$ measured by the SAMI-pH (*circles*) and interpolated from the spec-pH (*line*), corrected to bottle sample and CRM data; (*g*) NOAA/PMEL-measured TA and (*h*) DIC from bottle samples and normalized to salinity; (*i*) seawater $pCO_2$ and (*j*) saturation state of aragonite ($\Omega_{arag}$) calculated from interpolated nDIC and nTA data via CO2SYS; (*k*) the observed carbon uptake ratio (CAR) as ($nDIC_{exp} - nDIC_{control}$) / $\Delta TA_{NaOH\ addition}$ (*solid*) and the theoretical CAR (*dashed*) from a CO2SYS calculation using measured TA and the average $pCO2_{atm}$ to estimate the equilibrium change in DIC (*dashed*); (*l*) the percent equilibration estimated between the observed and theoretical CAR. Data gaps in panels *a*, *b*, and *e* are due to connectivity issues while offloading sensor data.

The $\Delta nTA$ and $\Delta nDIC$ values calculated between the control and experiment tanks are summarized in Figure 3 where nTA and nDIC were interpolated between bottle samples measured at NOAA-PMEL, and/or were calculated via CO2SYS using sensor $pH_T$ and TA measured at Stony Brook University corrected to less frequent NOAA-PMEL TA and DIC bottle samples. The ratio of the $\Delta nDIC$ to the addition of alkalinity in the form of NaOH, or $\Delta nTA$, is included in Figure 3 for all experiments except that of the bulk $pH_T$ increase to 10.3. Neglecting experiments that were terminated before full equilibration, the final observed CAR ranged from 0.75 ± 0.04 to 0.87 ± 0.08 (Table 1).

An anomalous event was noted in both the experiment and control cases for the target $pH_T$ 8.5 experiment with forced air movement across the surface of the tank, wherein an increase in TA and DIC was noted around day 30 of the experiment. The cause of this event is unclear but could include biological changes in both tanks, the introduction of alkalinity from environmental contaminants, or the anomalous delayed release of alkalinity from suspended solids. This event was not observed in any other case, and highlights the importance of using controls to understand complex interactions in these experiments. A time-series including this event is available in the Supplementary Materials.

Henry's law and CO2SYS calculations were used to estimate the initial and final equilibration condition of each tank experiment. LiCOR $pCO_{2,atm}$ measurements were averaged across experiments to a representative value of 421 ± 14 ppm, which was used with the initial seawater temperature and salinity to estimate $pCO_{2,seawater}$ at the beginning of each experiment. This initial $pCO_{2,seawater}$ was in all cases greater than the atmospheric $pCO_{2,seawater}$, indicating that the seawater was not fully equilibrated with the atmosphere at the time when NaOH was added, likely due to respiration and decomposition of organic material (Section 2.1), and as such, the tanks should outgas $CO_2$. The initial equilibrium DIC was estimated from a CO2SYS calculation using the $pCO_{2,seawater}$ and $nTA_i$, which in all cases was less than the initial $nDIC$ measured or calculated from $nTA_i$ and $pH_{T,i}$ (by 29 – 108 µmol kg$^{-1}$). These observations underscore the importance of having a control tank to capture natural dynamics of $CO_2$ ingassing and outgassing to ensure that changes in DIC attributed to OAE are correctly accounted for.

The final equilibrium $nDIC$ was estimated from a CO2SYS calculation using the same $pCO_{2,seawater}$ and the $nTA$ measured just after the NaOH addition, corrected for the linear increase in salinity over the course of the experiment. The ratio of the expected $\Delta nDIC$ calculated at equilibrium with the atmosphere to the addition of alkalinity provides a simple estimate of the expected $CO_2$ storage capacity for a given experiment. The percent equilibration for each experiment was then estimated from the measured and expected values for CAR. Within the series of experiments with a targeted $pH_T$ of 8.5, the timeline to reach an estimated 90% $CO_2$ equilibration decreased from 65 days (with internal circulation but still water at the surface of the tank), to 50 days (with the addition of forced air movement across the surface of the tank) to 22 days (with the addition of air bubbling). We note that only the two cases (targeted $pH_T$ of 8.5 and 9.5) with the addition of air bubbling reached full equilibration with the atmosphere.

**Table 1:** Range of variables measured, calculated, or extrapolated in large tank experiments, where M denotes direct
measurement, C denotes calculation via CO2SYS, and E denotes extrapolation to equilibrium conditions. Subscripts
*i* and *f* refer to initial and final conditions, and 'peak' refers to the time point immediately after the addition of
NaOH.

| pH target | - | 8.5 | | 8.5 | | 8.5 | | 8.7 | | 9.5 | | 10.3 | |
|---|---|---|---|---|---|---|---|---|---|---|---|---|---|
| Surface condition | - | Still | | Forced Air | | Forced Air and Air Bubbles | | Still | | Forced Air and Air Bubbles | | Still | |
| Tank (C = control, E = experiment) | - | C | E | C | E | C | E | C | E | C | E | C | E |
| $\Delta TA$ = NaOH addition ($\pm$ 10 $\mu$mol/kg) | M | 0 | 409 | 0 | 462 | 0 | 375 | 0 | 626 | 0 | 1406 | 0 | 3305 |
| Salinity$_i$ (g/kg) | M | 28.7 | 28.7 | 30.2 | 30.2 | 30.4 | 30.4 | 26.9 | 26.8 | 26.9 | 26.9 | 28.5 | 28.4 |
| Salinity$_f$ (g/kg) | M | 30.5 | 30.2 | 37.3 | 36.6 | 34.7 | 33.7 | 27.6 | 27.6 | 29.0 | 29.2 | 28.6 | 28.6 |
| $pH_{T,i}$ ($\pm$ 0.005) | M | 7.76 | 7.76 | 7.73 | 7.73 | 7.93 | 7.93 | 7.92 | 7.75 | 7.95 | 7.95 | 7.70 | 7.75 |
| $pH_{T,peak}$ ($\pm$ 0.005) | M | - | 8.54 | - | 8.58 | - | 8.49 | - | 8.68 | - | 9.51 | - | 10.10 |
| $pH_{T,f}$ ($\pm$ 0.005) | M | 7.88 | 8.05 | 7.85 | 7.99 | 7.99 | 8.01 | 7.84 | 8.26 | 8.01 | 8.21 | 7.75 | 9.52 |
| $nTA_i$ ($\pm$ 10 $\mu$mol/kg) | M | 2049 | 2049 | 2069 | 2069 | 2248 | 2248 | 2075 | 2075 | 2007 | 2007 | 2023 | 2025 |
| $nTA_{peak}$ ($\pm$ 10 $\mu$mol/kg) | M | - | 2458 | - | 2531 | - | 2623 | - | 2701 | - | 3414 | - | 5330 |
| $nTA_f$ ($\pm$ 10 $\mu$mol/kg) | M | 2080 | 2528 | 2235 | 2674 | 2246 | 2624 | 2095 | 2696 | 2014 | 3363 | 2041 | 1253 |
| $nDIC_i$ ($\mu$mol/kg) | M | 1944 | 1947 | 1957 | 1996 | 2082 | 2087 | 1897 | 1975 | 1852 | 1852 | 1928 | 1938 |
| $nDIC_f$ ($\mu$mol/kg) | M | 1908 | 2280 | 2084 | 2433 | 2027 | 2365 | 1937 | 2336 | 1832 | 2977 | 1947 | 720 |
| $\Omega_{aragonite,i}$ | C | 1.39 | 1.37 | 1.4 | 1.1 | 2.0 | 2.0 | 2.4 | 2.4 | 1.9 | 1.9 | 1.4 | 1.3 |
| $\Omega_{aragonite,peak}$ | C | - | 5.9 | - | 6.0 | - | 6.2 | - | 8.8 | - | 19.3 | - | 30.3 |
| $\Omega_{aragonite,f}$ | C | 2.0 | 3.0 | 1.7 | 2.8 | 2.5 | 3.0 | 1.9 | 4.4 | 2.1 | 4.9 | 1.4 | 5.2 |
| $CAR_f$ | C | - | 0.85 $\pm$ 0.04 | - | 0.75 $\pm$ 0.04 | - | 0.87 $\pm$ 0.08 | - | 0.52 $\pm$ 0.07 | - | 0.82 $\pm$ 0.09 | - | - |
| $CAR_{equilibrium}$ | E | - | 0.89 | - | 0.85 | - | 0.85 | - | 0.84 | - | 0.81 | - | - |
| % equilibration (time elapsed in days) | E | - | 95 $\pm$ 10 (67) | - | 92 $\pm$ 10 (63) | - | 102 $\pm$ 12 (45) | - | 79 $\pm$ 6 (42) | - | 104 $\pm$ 7 (74) | - | (13) |

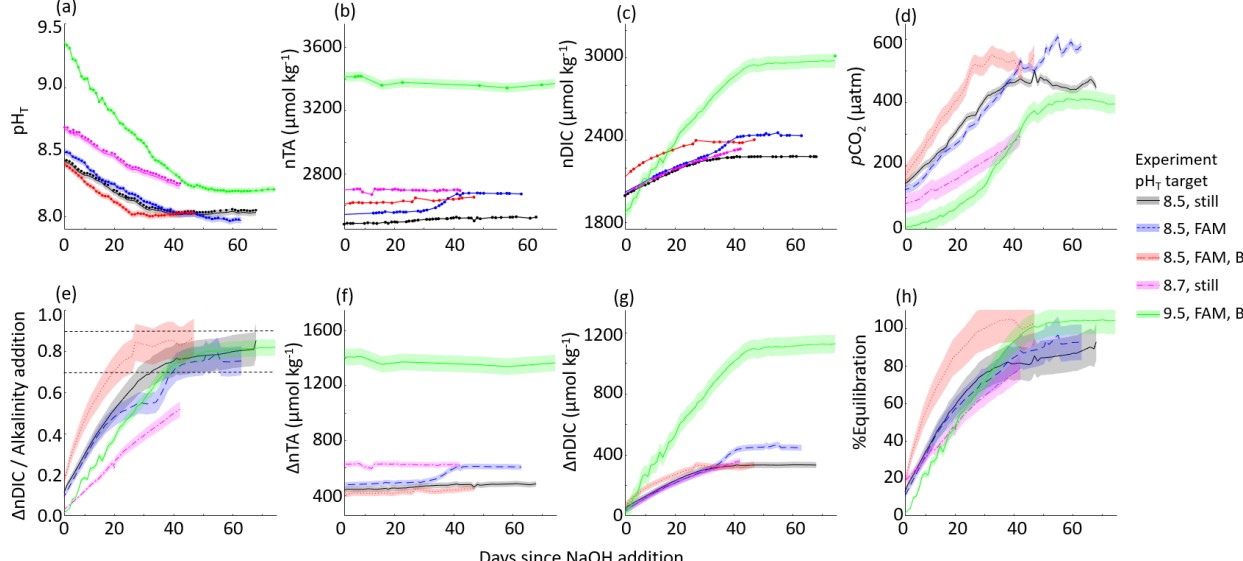


**Figure 3**: Results of 5 tank-scale experiments in which enough NaOH was added to each tank to raise the bulk $pH_T$ to 8.3 – 9.7. $pH_T$ decreased rapidly in all cases in which air bubbling sped equilibration with atmospheric $CO_2$. Results include: (a) measured $pH_T$, (b) measured nTA, (c) measured nDIC or CO2SYS calculated (for $pH_T$ 9.5 case only), (d) CO2SYS -calculated $pCO_2$, (e) the observed carbon uptake ratio (CAR) as ($nDIC_{exp}$ - $nDIC_{control}$) / $\Delta nTA_{NaOH\ addition}$ with horizontal dashed lines representing the expected range of 0.7-0.9 mol $CO_2$ uptake / mol NaOH added to seawater, the change in (f) nTA and (g) nDIC compared to the baseline measurements before the addition of NaOH, and the percent equilibration estimated between the observed and theoretical CAR.

### 3.2 Aquaria experiments

Table 2 provides a summary of the range of oceanographic variables quantified for the aquaria experiments.

**Table 2:** Range of variables measured, calculated, or extrapolated in aquaria experiments, where M denotes direct measurement, C denotes calculation via CO2SYS, and E denotes estimation within specified equilibration conditions. Subscripts *i* and *f* refer to initial and final conditions, and 'peak' refers to the time point immediately after the addition of NaOH.

| pH target | - | 0 Control | 8.3 | 8.5 | 8.5 Without air bubbles | 8.7 | 9.3 | 9.5 | 9.7 | 9.9 | 10.0 | 10.1 | 10.2 | 10.3 |
|---|---|---|---|---|---|---|---|---|---|---|---|---|---|---|
| $\Delta TA$ = NaOH addition (± 10 µmol/kg) | M | 0 | 187 | 331 | 362 | 543 | 1409 | 1679 | 2037 | 2216 | 2276 | 2504 | 2796 | 3829 |
| $pH_{T,i}$ (± 0.1) | M | 7.94 | 7.97 | 7.90 | 7.86 | 7.95 | 7.98 | 7.98 | 7.98 | 8.06 | 8.04 | 8.04 | 8.04 | 7.95 |
| $pH_{T,peak}$ (± 0.1) | M | - | 8.28 | 8.41 | 8.40 | 8.63 | 9.22 | 9.43 | 9.64 | 9.83 | 9.91 | 10.23 | 10.32 | 10.20 |
| $pH_{T,f}$ (± 0.1) | M | 8.06 | 8.03 | 8.07 | 8.11 | 8.08 | 8.21 | 8.20 | 8.23 | 8.65 | 8.96 | 8.72 | 9.46 | 7.99 |
| $TA_i$ (± 10 µmol/kg) | M | 2265 | 2262 | 2250 | 2250 | 2250 | 2393 | 2393 | 2393 | 2531 | 2531 | 2531 | 2531 | 2250 |
| $TA_{peak}$ (± 10 µmol/kg) | M | - | 2449 | 2582 | 2611 | 2793 | 3801 | 4072 | 4430 | 4748 | - | - | - | 4608 |
| $TA_f$ (± 10 µmol/kg) | M | 2323 | 2476 | 2640 | 2645 | 2822 | 3837 | 4110 | 4420 | 4462 | 1702 | 1835 | 1537 | 2202 |
| $DIC_i$ (µmol/kg) | C | 2089 | 2073 | 2091 | 2107 | 2070 | 2192 | 2192 | 2192 | 2282 | 2287 | 2287 | 2287 | 2067 |

| | | | | | | | | | | | | | | |
|---|---|---|---|---|---|---|---|---|---|---|---|---|---|---|
| $DIC_f$ (μmol/kg) | C | 2113 | 2246 | 2377 | 2382 | 2540 | 3372 | 3486 | 3877 | 3389 | 992 | 1244 | 671 | 2003 |
| $\Omega_{aragonite,i}$ | C | 2.1 | 2.2 | 1.9 | 1.8 | 2.1 | 2.34 | 2.4 | 2.4 | 2.9 | 2.8 | 2.8 | 2.8 | 2.1 |
| $\Omega_{aragonite,peak}$ | C | - | 4.2 | 5.5 | 5.5 | 8.1 | 19.5 | 23.1 | 27.0 | 29.8 | 30.2 | 30.9 | 32.4 | 38.9 |
| $\Omega_{aragonite,f}$ | C | 2.4 | 2.7 | 3.1 | 3.1 | 3.4 | 5.9 | 7.9 | 7.1 | 13.7 | 6.5 | 5.7 | 7.0 | 2.2 |
| $CAR_f$ | C | - | 0.92 ± 0.10 | 0.87 ± 0.06 | 0.76 ± 0.05 | 0.87 ± 0.04 | 0.84 ± 0.02 | 0.86 ± 0.02 | 0.84 ± 0.02 | 0.50 | - | - | - | - |
| $CAR_{equilibrium}$ | E | - | 0.69 | 0.67 | 0.64 | 0.77 | 0.80 | 0.80 | 0.80 | 0.81 | - | - | - | - |
| % equilibration (time elapsed in days) | E | (40) | 130 (16) | 126 (18) | 116 (40) | 111 (16) | 104 (18) | 106 (18) | 104 (18) | 62 (1) | (1) | (1) | (1) | (16) |
| $CaCO_3$ precipitation? | M | - | No | No | No | No | No | No | No | No | Yes | Yes | Yes | Yes |

The aquaria experiments are not directly comparable to the control stated in Table 2. Seawater for one control
aquarium was collected in March 2023, and was monitored for $pH_T$ and TA changes through May 2023. Seawater
for the experimental aquaria was collected in three batches between March, April, and May 2023, with only 4-6
aquaria experiments running in parallel within each set of experiments due to space and analytical throughput
constraints. Because of this, the experiments started in March 2023 could be compared directly to the control (target
pHT 8.3, 8.5, 8.5 still, and 8.7), but the rest of the experiments used different initial seawater than the control
aquarium. The CAR for each aquaria experiment was therefore calculated from changes in DIC and TA between the
initial 'baseline' condition and after the NaOH was added within a given aquarium, rather than between the
experiment and control cases. The CAR ranged between 0.76 ± 0.05 and 0.92 ± 0.10, excluding cases where mineral
precipitation was evident and for the $pH_T$ 9.9 case where the experiment ended after one day due to a sensor logging
failure. This wide range in ΔDIC/ ΔTA is likely due to the limited number of TA samples collected throughout these
experiments (daily at best with no duplicates due to the limited volume), and the imprecision of electrode-based $pH_T$
measurements relative to the SAMI-pH and spec-pH based measurements used in the large tank experiments.
No significant changes in salinity were recorded during these experiments as measured by a handheld salinometer
with a range of 30 - 31. Therefore, DIC and TA were not normalized to salinity. Temperature values ranged from 19
- 21 °C during the experiments.
Similar to the large tank experiments, we used Henry's law and CO2SYS calculations to estimate the initial and
final equilibration condition of each aquaria experiment. The same average $pCO_{2,atm}$ of 421 ± 14 ppm was used with
the initial seawater temperature and salinity to estimate $pCO_{2,seawater}$ at the beginning of each experiment. The initial
equilibrium DIC was estimated from a CO2SYS calculation using this $pCO_{2,seawater}$ and $TA_i$, which in all cases was
less than the initial DIC calculated from $TA_i$ and $pH_{T,i}$ (by $16 - 36$ μmol kg$^{-1}$). This indicates that the seawater was
not fully equilibrated with the atmosphere at the time when NaOH was added, likely due to respiration and
decomposition of organic material (Section 2.1), and as such, the aquaria should outgas $CO_2$. The final equilibrium
DIC was estimated from a CO2SYS calculation using the same $pCO_{2,seawater}$ and the TA measured just after the
NaOH addition. The percent equilibration for each experiment was then estimated between the measured and
predicted values for ΔDIC/ ΔTA. Due to the air bubbling, most experiments approached equilibrium with the
atmosphere within 1-7 days, with the exception of the non-bubbled $pH_T$ 8.5 experiment that took ~20 days. The
surface water of this non-bubbled experiment was stagnant, and the water was only mixed via stirring just before
taking pH and TA samples. Absorption of atmospheric $CO_2$ began immediately after the NaOH addition, as
determined by decreasing $pH_T$. We note that there are significant uncertainties in these equilibrium estimates leading
to estimates of >100% equilibration. These estimates would be better constrained with more continuous carbonate
chemistry measurements, particularly seawater and atmosphere $pCO_2$ throughout the experiments that would allow
for more direct calculation of air-sea $CO_2$ flux and equilibration, and finer control of bubbling and diffusion rates are
necessary to define the timeline for equilibration within the aquaria.
Each aquarium was gently stirred during the addition of NaOH to prevent and/or redissolve $Mg(OH)_2$ precipitation.
No $CaCO_3$ precipitation was observed in the tanks below a bulk seawater $pH_T$ of 10.0, and TA remained stable in
each of these experiments with the exception of some increase driven by minor evaporation on the order of +2
umol/kg per day. Experiments where $CaCO_3$ precipitation was induced by increasing the starting pH to values above
10 are discussed in Section 3.3.
The aquaria experiments with target $pH_T$ from 8.3 – 9.9 are summarized in Figure 4.

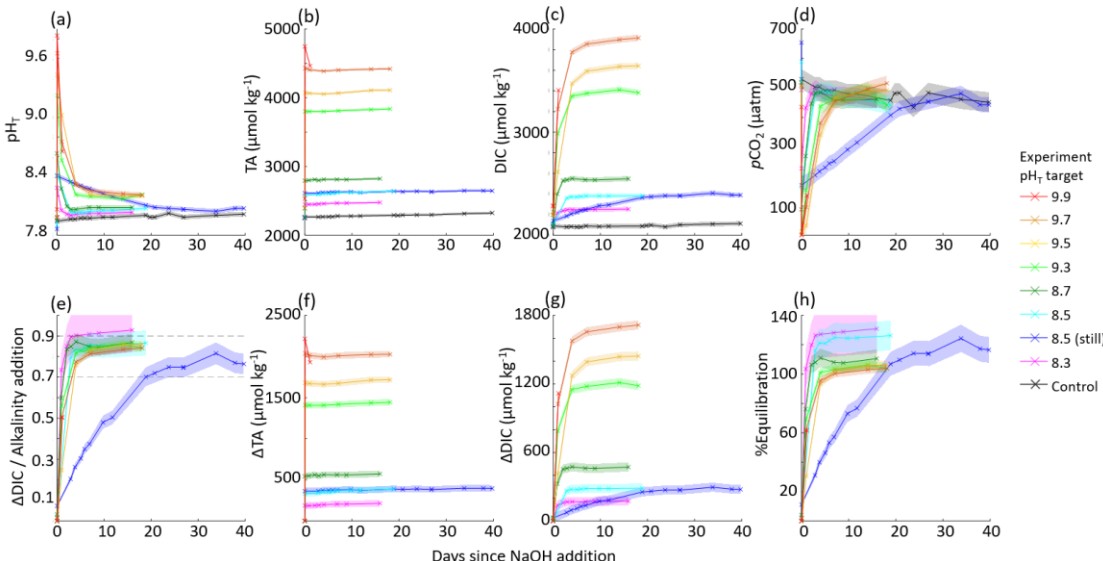


**Figure 4:** Results of 9 aquaria experiments in which enough NaOH was added to each aquaria to raise the bulk $pH_T$
to 8.3 – 9.9. $pH_T$ decreased rapidly in all cases in which air bubbling sped equilibration with atmospheric $CO_2$.
Results include: (a) measured $pH_T$, (b) measured TA, (c) CO2SYS-calculated DIC, (d) CO2SYS-calculated $pCO_2$,
(e) the observed carbon uptake ratio (CAR) as $(DIC_{exp} - DIC_{baseline}) / \Delta TA_{NaOH\ addition}$ with horizontal dashed lines
representing the expected range of 0.7-0.9 mol $CO_2$ uptake / mol NaOH added to seawater, the change in (f) TA and
(g) DIC compared to the baseline measurements before the addition of NaOH, and the percent equilibration
estimated between the observed and theoretical CAR.
In general, the large tanks and aquaria showed reasonable agreement in achieving values for CAR within the
expected range of 0.7-0.9 (He and Tyka, 2023; Burt et al., 2021; Wang et al., 2023). While the use of aquaria
bubbled with air to speed equilibration allowed for a greater range of data collection within a constrained experiment
timeline, the quality of this data is limited by the lack of appropriate sensors to fit into these small 15 L aquaria and
fewer bottle samples due to the reduced quantity of seawater. However, while the large tanks allow for a larger
range of oceanographic sampling and sensing techniques, it is more challenging to fully quantify mixing and
circulation rates in the current large tank experimental setup.
Figure 5 shows the dependence of the equilibrium values of $\Delta DIC$, CAR, and $\Delta pH_T = (pH_{final} - pH_{initial})$ as a function
of the alkalinity addition for both tank and aquaria experiments in which the final percent equilibration for $CO_2$ was
estimated at greater than 90%. Results for tank and aquaria experiments aligned well, with increasing $\Delta DIC$ for
increasing alkalinity additions. The CAR was observed for all experiments to fall within the range expected for
seawater with the temperature and salinity values used in these tests. As expected from calculations of the response
of the seawater carbonate buffer system to additions of alkalinity, the $pH_T$ at equilibrium exceeded the initial $pH_T$
value prior to the addition of alkalinity. That is, even once equilibrium in the alkalinity enhanced experiment tank
had been reached, the ending pH value was slightly elevated relative to the starting pH value. This finding warrants
further investigation on the potential of OAE to mitigate some acidification impacts in controlled field trials by
metering the discharge of alkalinity to semi-protected water body.

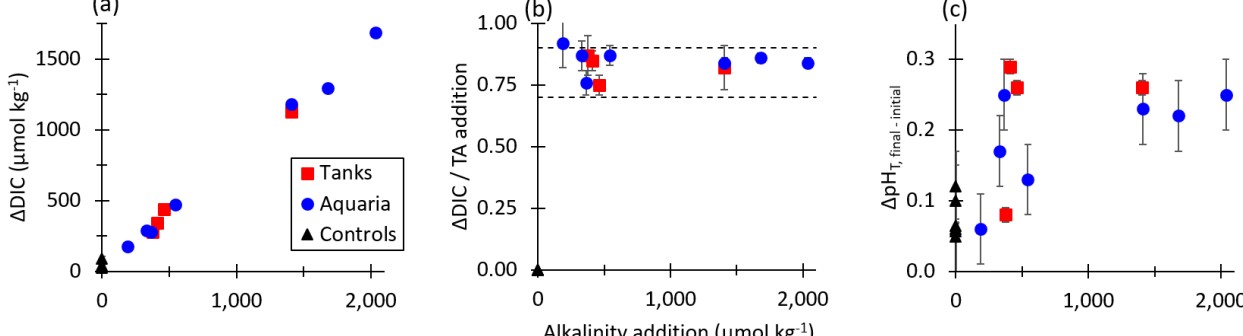

**Figure 5**: (a) The change in final CO2SYS-predicted DIC relative to the initial conditions for tank, aquaria, and
control experiments increases with increasing NaOH additions for cases where the air-sea $CO_2$ equilibration was
estimated at >90% at the termination of each experiment. (b) CO2SYS-predicted CAR ($\Delta$DIC / Alkalinity addition)
at air-sea equilibrium conditions for tank, aquaria, and control experiments, with horizontal dashed lines
representing the expected range of 0.7-0.9 mol $CO_2$ uptake / mol NaOH added to seawater. (c) The measured $\Delta pH_T$
= ($pH_{final}$ – $pH_{initial}$) increases with
alkalinity addition for both tank and aquaria experiments.

## 3.3 Experiments exceeding the CaCO₃ precipitation threshold

While $Mg(OH)_2$ precipitation occurs immediately upon introduction of concentrated (i.e., ~0.5 M) NaOH to still
seawater, it may be rapidly dissolved or avoided entirely by gentle mixing, including via the use of stirrers,
circulation pumps, or air bubblers. This precipitation and redissolution happened rapidly enough that it was not
identified in any TA or other variables measured in the aquaria and tank tests. However, in cases where enough
NaOH was added to raise the bulk seawater $pH_T$ to greater than 10.0 (i.e. in one large tank test with a target $pH_T$ of
10.3, and in 4 aquaria experiments ranging from $pH_T$ 10.0-10.3), runaway precipitation of $Mg(OH)_2$ and $CaCO_3$ was
observed. This was characterized by a sharp reduction in both TA and DIC and an increase in turbidity. Runaway
precipitation has been described as a condition in which more alkalinity is removed from seawater by mineral
precipitation than was initially added until a new steady state is achieved (Moras et al., 2022; Hartmann et al., 2023;
Suitner et al., 2023). This can significantly impact the efficiency of OAE, and has implications for biological
productivity, as increased turbidity may impact photosynthesis or predator-prey interactions.
In both the tank and aquarium $pH_T$ 10.3 cases, discrete samples of the precipitate were collected at seven different
times after the bulk $pH_T$ value reached 10.3 (0h, 3h, 24h, 49h, 71h, 145h, 167h - see Fig. 6) for XRD and SEM
analysis. At each timepoint, 0.5 – 1 L seawater was collected from the tank sampling port or from the center of the
aquaria. In cases where precipitation had visibly settled at the bottom of the aquaria, this material was stirred into the
water column before sampling. We note that material that settled to the bottom of the large tanks was not directly
collected, and that only a subset of precipitation was collected at each time point, such that later timepoints may
include solids that had precipitated at the beginning of the experiment. The filtered seawater was immediately
analyzed for TA and pH via Ross electrode because the heightened pH was out of the range of spectrophotometric
methods. Bottle samples of filtered seawater were not able to be analyzed at NOAA PMEL due to the continued
precipitation of $CaCO_3$ after filtration and preservation. Both XRD and SEM results showed the dominance of
$Mg(OH)_2$ precipitation immediately after the alkalinity addition and the corresponding increase in pH and $\Omega_{aragonite}$
(to a value of around 30). The $Mg(OH)_2$ precipitation at this stage was thick, slurry-like, and difficult to
appropriately rinse. Broad peaks associated with brucite in the 0 and 3 hr time points may reflect that these signals
were partially obscured by the presence of other salts, and a sharp peak in the 0 hr time point ~27° 2θ is likely
associated with NaCl. Within hours of the NaOH addition, the runaway CaCO₃ precipitation began, characterized by
fine, light particulates in the water column and a sharp increase in turbidity. Within ~24 hours of the NaOH addition,
most Mg(OH)₂ signals had disappeared, leaving only aragonite and calcite peaks in the XRD. The results of the
XRD analysis for the tank experiment are summarized in Figure 7, and the aquarium experiment showed similar
results (see Supplementary Materials). TA decreased throughout the precipitation of Mg(OH)₂ and CaCO₃, and was
below that of the initial seawater within 24 hours of the NaOH addition. In the tank experiment, the initial TA (2025
μmol/kg) was raised by 3305 μmol/kg. Within 3 days the TA had decreased to 1583 μmol/kg and continued to
decrease through the termination of the experiment to 1253 μmol/kg 10 days after the addition of NaOH. The DIC,
which was initially measured at 1938 μmol/kg, decreased to 720 μmol/kg by the end of the experiment. This
experiment shows that runaway CaCO₃ can result in a significant loss of both efficiency of alkalinity dosing for
OAE projects and of storage of carbon in the form of DIC. A figure of time-series data collected during the tank
experiment is available in the Supplementary Materials.

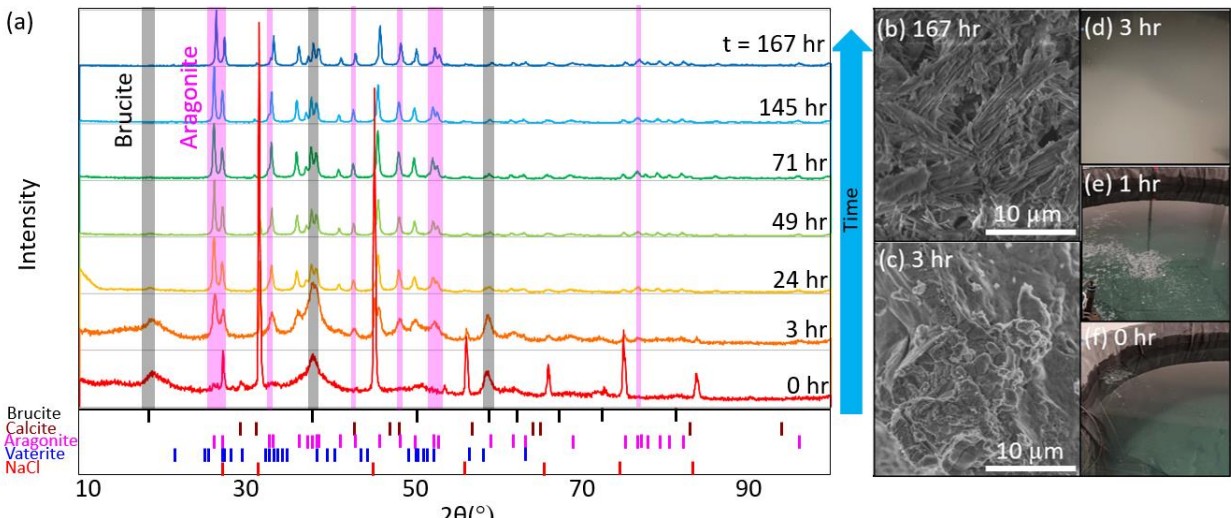


**Figure 6:** (*a*) XRD analysis of particulates filtered from seawater after the addition of enough NaOH to raise the
bulk seawater pH$_T$ to 10.3 showed mineral precipitation initially dominated by Mg(OH)₂ before it was overtaken by
CaCO₃,arag. The shaded grey vertical bars highlight several peaks characteristic of brucite which typically disappear
after 24 hours, and the shaded blue bars represent several aragonite peaks which appear between 3 and 24 hours.
Representative SEM images show (*b*)CaCO₃,arag at the end of the experiment, and (*c*) Mg(OH)₂ captured ~3 hours
after the NaOH addition. Photographs of the tank experiment show seawater (*d*) ~3 hours after the NaOH addition,
when runaway CaCO₃ precipitation became visually apparent, (*e*) during NaOH addition into still water (i.e.,
without the use of stirrers, circulation pumps, or air bubblers to break up and redissolve Mg(OH)₂), and (*f*) before
NaOH addition.
In summary, the presence and duration of brucite precipitation upon addition of 0.5 M aqueous NaOH depends on
the ratio of the NaOH addition rate to the local dilution rate in the receiving waters. Future research using flow
through tanks could help identify thresholds below which brucite precipitation can be avoided or limited, and the
immediate formation of Mg(OH)₂ may be reversible, as also noted by Suitner et al. (2023) and Cyronak et al.
(2023). At the given initial seawater conditions, the threshold for aragonite precipitation began at an $\Omega_{arag}$ of 30,
corresponding to pH$_T$ > 10.0, and continued as $\Omega_{arag}$ decreased to ~5.2. This threshold corresponded to an increase in
TA of >2270 μmol/kg. The potential for runaway aragonite precipitation may be reduced by active mixing at the
point of NaOH introduction, maintaining a mixing volume below bulk seawater pH$_T$ of 10.0, and allowing for
appropriate dilution in flow-through conditions, particularly on timescales of hours after alkalinity addition.
Additional characterization of runaway precipitation thresholds at varying temperatures, salinities, and suspended
particulate conditions will allow for OAE implementation designs that best avoid this potential risk to OAE
efficiency and ecosystem perturbation. We note that these results are only valid for experiments that are open to the
atmosphere allowing for exchange of $CO_2$ across the air-sea boundary using an aqueous hydroxide feedstock for
alkalinity, and are not comparable to experiments such as closed bottle incubations, where sustained conditions of
high $\Omega_{arag}$ may result in precipitation at different thresholds. We also note that we do not assume zero aragonite
precipitation at conditions below the stated thresholds, but that potential precipitation is not readily detectable with
our experimental setup. For example, heterogeneous $CaCO_3$ precipitation events, such as may occur on suspended
sediments in the water column, have been suggested through characteristic changes in seawater TA/DIC ratios in
cases of riverine inputs and bottom sediment resuspension (Bustos-Serrano et al., 2009; Wurgaft et al., 2016; 2021).
Suspended sediments in the context of OAE project sites could influence OAE efficiency and the potential for
runaway precipitation and should be included in future studies (Bach, 2023). The thresholds determined in this study
are significantly higher than those of some mineral-based OAE studies, including precipitation after an increase in
TA of ~500 μmol/kg using CaO and $Ca(OH)_2$ mineral additions (Moras et al., 2022). Hartmann et al. (2022) noted
precipitation resulting from alkalinity additions of >600 μmol/kg $Mg(OH)_2$, and found that alkaline solutions
avoided carbonate precipitation better than reactive alkaline particle additions to seawater. Suitner et al. (2023)
suggested that alkalinity additions > 2000 μmol/kg could be achievable given sufficient dilution to avoid runaway
precipitation. Together, these studies highlight the need to expand research into runaway precipitation to
characterize potential inefficiencies in OAE, particularly in in-situ experiments to establish relationships applicable
to ocean environments.
**5 Summary and Future Work**
These results demonstrate that ocean alkalinity enhancement using aqueous sodium hydroxide in seawater results in
$CO_2$ removal from air at an efficiency of 0.75 (± 0.04) – 0.92 (± 0.10), with 90% equilibration typically achieved
within 7 - 9 weeks (still surface water with ~16 L/min subsurface circulation through UV arrays) to 3 - 5 weeks
(with the addition of ambient air bubbling into the bottom of each tank) of the initial addition when performed in
~6000 L tanks with seawater-air contact areas of around 4.6 $m^2$. These results are in general agreement with ratios
noted in Burt et al. (2021), He and Tyka (2023), and Wang et al., (2023). Here, uncertainties are driven by sensor
precision and temporal resolution in discrete DIC and TA sampling, the limited number of experiments with
minimal opportunities for duplicates or replicates, and poorly constrained data on mixing, circulation, and air
bubbling rates. Ongoing experiments seek to improve each of these conditions and should particularly focus on
constraining the movement of water within a given tank to improve air-sea equilibration estimates and to allow for
better extrapolation from tank to field experiments. In addition, a focus of ongoing and future work is to provide rate
estimates for the uptake of atmospheric $CO_2$ in response to an NaOH addition, allowing for fitting and extrapolation
of a shortened experiment to equilibration with the atmosphere.
We relied on several methods to constrain seawater carbonate chemistry. The tank-scale experiments primarily
relied on discrete (at most daily) DIC and TA sampling (NOAA PMEL), paired with daily measurements from
spectrophotometric pH systems (SAMI-pH and a semi-automated benchtop spec-pH system following Carter et al.
(2013)) and local TA measurements. With appropriate calibration or correction of the spec-pH systems relative to
CRM, there was no significant difference in carbonate calculations using the NOAA PMEL DIC-TA or spec-pH-
local TA pairings, though the latter case typically produced larger uncertainties. Aquaria experiments relied on a
standard glass pH electrode (at most daily, corrected to spectrophotometric systems) with discrete (at most daily) TA
measurements, which provided reasonable data relative to the tank experiments. As a result, ongoing tank-scale
experiments have limited the volume of discrete DIC and TA samples collected for analysis at NOAA PMEL to
allow for faster and less expensive monitoring via spec-pH and local TA titrations. However, we note that the major
limitation in this measurement pathway lies in the spec-pH method, which is typically limited to $pH_T$ measurements
ranging from 7 – 9 for the meta-cresol purple indicator dye used. While our measurements retained some sensitivity
up to $pH_T$ 9.5, such a method should typically be considered unreliable at these $pH_T$ values, and we relied on
frequent correction to CRM and comparison with DIC/TA samples. Thymol blue is an alternative
spectrophotometric $pH_T$ indicator dye with sensitivity over the higher $pH_T$ conditions observed during these initial
trials and will be assessed for future experiments (Zhang and Byrne, 1996; Liu et al., 2006).
Aqueous NaOH with concentrations as high as 0.5 M can be added directly to turbulent seawater with only limited
observable precipitation of $Mg(OH)_2$. In these conditions this precipitated mineral rapidly redissolves on the
timescales of minutes to seconds. Improved control over the NaOH dosing rate (in our tank experiments, ~50 mL
NaOH/min) and the turbulence of the receiving water through metered flow through experiments will be valuable in
extrapolating to field conditions. This precipitation is detectable both visually and through turbidity measurements
and implies that straightforward measurement of pH and turbidity at the dispersal site can be used to adjust the
alkalinity dispersal rate according to local mixing conditions such that $Mg(OH)_2$ precipitation is avoided and/or
redissolves when it occurs. No significant $CaCO_3$ precipitation was observed at pH <10.0 or $\Omega_{aragonite}$ < 30.0.
Runaway $CaCO_3$ precipitation was observed above these thresholds, where a massive precipitation and settling of
$Mg(OH)_2$ and $CaCO_3$ solids results in less alkalinity in the overlying water than at the starting condition. pH and
turbidity sensing combined with discrete TA measurements could be used as a feedback signal for alkalinity dosing
into seawater to ensure that the local maximum thresholds at the dispersal location do not approach or exceed
conditions that promote significant $CaCO_3$ precipitation. We note that future investigations seeking to better
approximate field conditions should take into account seasonal and tidal shifts in temperature and salinity, and
varying conditions of suspended sediment in the water column, including that of aerial dust, terrestrial runoff, and
resuspended bottom sediments.
In these experiments, the seawater was filtered and bleach treated prior to experiments to limit biological growth,
and both tank and aquaria experiments were conducted indoors with limited light. Nevertheless, in most
experiments, biological growth was observed after a few weeks, including cyanobacteria and coccolithophores. A
series of experiments are underway to test the difference in $CO_2$ removal efficiency for two side-by-side tanks, both
of which are dosed with NaOH, but only one of which was bleached. Preliminary results show minimal difference
between the bleached and unbleached tanks, indicating these experiments are applicable to real-world conditions, at
least for regions with biological communities similar to that of Long Island Sound, but further investigation is
warranted.
A focus of future work is to consider the potential impact of the SEAMATE process on local ocean acidification
mitigation efforts. We note that in each constrained tank and aquarium experiment, the $pH_T$ at equilibrium exceeds
the initial $pH_T$ value prior to the addition of alkalinity. A controlled release of alkalinity could theoretically be
configured to maintain a locally elevated $pH_T$ value relative to pre-alkaline conditions, with potential uses in
aquaculture and hatchery environments.
These results provide clear and practical guidelines for MRV for OAE implementations employing aqueous
alkalinity. First, carbonate chemistry and turbidity measurements at the alkalinity dispersal location can ensure that
seawater parameters such as pH and $\Omega_{aragonite}$ remain within pre-determined safe bounds and that unwanted
precipitation is avoided. Second, for a given OAE deployment, where ocean models provide a reasonable certainty
about the fraction of the alkalinity plume remaining in the surface over weeks to months, the $CO_2$ removal efficiency
and timescale for air-seawater equilibration provided by our experiments can place a lower bound on the amount of
$CO_2$ removal expected from that OAE intervention. Expanding these studies from tank scale to mesocosm and field
experiments will be crucial to understanding biological impacts and constraining realistic air-sea interactions in
response to this type of OAE (Oschlies et al., 2023).
**Data availability**
Data are described in the manuscript and provided Supplementary Materials, which includes a .csv file with
processed sensor and sample time-series data at hourly resolution.
**Author contribution**
MDE and BRC designed the experiments and MCR carried them out with support from NH, CS, and XL. JH
provided support on experimental setup and instrumentation. MCR prepared the manuscript with contributions from
all co-authors.
**Competing interests**
MCR is Lead Oceanographer and Head of MRV at Ebb Carbon, Inc.  MDE is Co-Founder and Chief Scientific
Advisor at Ebb Carbon, Inc.
**Acknowledgements**
We would like to than Stephen Abrams and Thomas Wilson at Stony Brook University Flax Pond Marine Lab for
technical assistance in experiment setup. We thank Chris Ikeda and Susan Curless of NOAA PMEL for support in
discrete sample analysis. We thank Mike Tyka for productive discussions. We thank Eyal Wurgaft for assistance in
TA titrations.
**Funding**
We acknowledge funding from The Grantham Foundation for the Protection of the Environment under the SEA
MATE (Safe Elevation of Alkalinity for the Mitigation of Acidification Through Electrochemistry) grant. In
addition, this research used the XRD facility of the Center for Functional Nanomaterials (CFN), which is a U.S.
Department of Energy Office of Science User Facility, at Brookhaven National Laboratory under Contract No. DE-
SC0012704.  BRC and JH were funded through the Cooperative Institute for Climate, Ocean, and Ecosystem
Studies (CICOES) under NOAA Cooperative Agreement NA20OAR4320271 and supported by NOAA's PMEL.

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
