# Peer review of "A comprehensive assessment of electrochemical ocean alkalinity"

_EGUsphere, 2024_

## Author Response (AR1)

To the reviewers and editors:

Thank you for taking your valuable time to carefully read the initial manuscript of "A comprehensive assessment of electrochemical ocean alkalinity enhancement in seawater: kinetics, efficiency, and precipitation thresholds" and for your thoughtful comments. This feedback has led to important improvements that we have incorporated into the revised manuscript.

Below in our Response to Reviewers, we address, point by point, each comment of the reviewers on the initial manuscript and carefully explain the changes that were made in response.

On behalf of all the coauthors,

Sincerely,

Dr. Mallory Ringham

Adjunct, Stony Brook University

Lead Oceanographer, Ebb Carbon, Inc.
* * *
**Responses to reviewer comments**

In the responses, "[Line N]" indicates that the location is line number N in the original version of the manuscript. We have made edits to the manuscript while working through the reviews, and reference the new text throughout these comments.

Link to original preprint: https://egusphere.copernicus.org/preprints/2024/egusphere-2024-108/
* * *
**RC1: Comment on egusphere-2024-108, Anonymous Referee #1, 29 Feb 2024**

**R1C1 (Referee 1, Comment 1):** The manuscript pretends to comprehensively assess the effectivity of the electrochemical OAE approach SEA MATE, but first confuses the targets of mitigating acidification, i.e. the Mitigation of Acidification Through Electrochemistry (MATE) part of the process acronym, and the removal of CO2 from the atmosphere, and, more importantly, is not about electrochemical OAE at all! It describes experiments in which NaOH is added to seawater. NaOH could, in theory, be produced electrochemically, but that does not even seem to be the case here. The manuscript describes what happens if you add NaOH to filtered and sterilized seawater in indoor tanks of different size.

**R1R1 (Referee 1, Response 1):** We appreciate the time the Referee spent reviewing our manuscript, but we have to take issue with both the content and the tone of this comment. There are multiple points addressed in this comment so we will address them one at a time:

**R1R1a -** First regarding the comment "The manuscript pretends to comprehensively assess the effectivity of the electrochemical OAE approach SEA MATE" It is clear from this comment and one of the referees later comments that our title indicated to them that our paper included an assessment of the electrochemical process for generating the alkalinity in addition to the effect of the alkalinity addition on CO2 removal. We can understand this confusion. The scope of the manuscript is limited to an assessment of the response of the ocean atmosphere system to the enhancement of alkalinity in the form of aqueous sodium hydroxide. Aqueous sodium hydroxide is the form of alkalinity generated by electrochemical ocean alkalinity approaches, hence the original title. However we understand in retrospect how this title could lead the reader to assume we are assessing the effectiveness of the electrochemical process that generates the alkalinity. As a result we have revised the title to "An assessment of ocean alkalinity enhancement using aqueous hydroxide: kinetics, efficiency, and precipitation thresholds."

**R1R1b -** Regarding the comment "...SEA MATE, but first confuses the targets of mitigating acidification, i.e. the Mitigation of Acidification Through Electrochemistry (MATE) part of the process acronym, and the removal of CO2 from the atmosphere..." We do not confuse these targets at all. This manuscript is about measuring the CO2 removal effects of alkalinity enhancement, that is the Safe Enhancement of Alkalinity, SEA, the first part of the process acronym. As we point out in the manuscript, the SEA MATE process results in a rapid chemical shift that mitigates several key metrics for ocean acidification (the "MATE"). These shifts in OA metrics are later mostly countered through enhanced CO2 uptake and storage (the "SEA") on a timescale of ~1-12 months. With transport and ocean mixing diluting the NaOH plume and moving it away from the NaOH release site, this differential in the time scale of the two processes can create a situation where the transient mitigation is sustained locally in a quasi-steady state throughout the duration of the NaOH release while still generating the expected amount of mCDR over longer time horizons. These steady state conditions may be tuned to mitigate ocean acidification by modulating the location and quantity of the release or by choosing an environment with limited exchange with the open ocean such as an embayment. This could be beneficial for shellfish farmers, for example. This is explained in the original manuscript at lines 130-140.

As we point out in the manuscript, even after full equilibration with the atmosphere has occurred the pH is somewhat higher than when it started, but any kind of global mitigation of ocean acidification is unfeasible because it would require amounts of CO2 removal beyond what is needed to mitigate climate change. To make this more clear and distinguish from the application of local ocean acidification mitigation, we have removed lines 130-140 from the original manuscript to now redirect the focus of the paper to the scope included.

**R1R1c-** Regarding the comment, "and, more importantly, is not about electrochemical OAE at all!". It is indeed - it is about the response of the ocean atmosphere system to electrochemical OAE in that electrochemical OAE typically produces alkalinity in the form of aqueous hydroxide. But the sense in which the referee means this comment, we suspect, is that it is not about the details of the electrochemical process itself. To avoid this misunderstanding, we have changed the title of the revised manuscript to "An assessment of ocean alkalinity enhancement using aqueous hydroxides: kinetics, efficiency, and precipitation thresholds" and have also deleted lines 72-90 and Fig. 1 from the original manuscript that described the details of a specific electrochemical process to generate alkalinity. We intend these changes to make clear the true intended scope of the manuscript.

**R1R1d-** Regarding the comment "The manuscript describes what happens if you add NaOH to filtered and sterilized seawater in indoor tanks of different size." This statement is factually true but the wording

implies that the studies we describe have limited applicability to actual OAE deployments. The experiments we described are directly relevant to Measurement, Reporting, and Verification (MRV) for real world ocean alkalinity enhancement implementations. Techniques that generate NaOH from seawater have been advanced as potential methods for OAE based mCDR. The process for evaluating these methods are necessarily cautious and stepwise. The fundamental chemistry for this process is known on a small scale, but the goal of this proof of concept study is to test whether the OAE method results in the expected mCDR when implemented on a large batch of precisely-monitored natural seawater. This scale of experiment is a critical step because it is among the largest possible experiments that allow for a plausible comparison of mCDR with and without alkalinity addition to natural seawater, as well as some plausible control of biological factors. For example, the reason that we bleach, or sterilize, the tanks prior to the experiments is to first separate out the effect of OAE from the effect of its interaction with biological growth.  Separate experiments that are now ongoing and will be published in the future are quantifying the effect of different forms of biological growth on OAE efficiency and kinetics in a systematic and controlled manner. As we describe in the manuscript and in our response to the referees, there is a reason for using different size tanks. The large tanks allow us to routinely subsample the seawater for the needed carbonate chemistry measurements at the required precision to evaluate this method. They also allow for deployment of larger oceanographic sensors than can be submerged in smaller experiments. The smaller aquaria are primarily employed to allow us to generate multiple replicate measurements of the CO2 removed to alkalinity ratio, that is the OAE efficiency.

Because the time scale for CO2 equilibration is typically longer than the time scale for mixing in the ocean, MRV for OAE will necessarily include the use of ocean models to quantify the amount of CO2 removed. To ground truth these models, we must have an understanding of the equilibrium value of the ratio of CO2 removed to alkalinity added under different conditions, and also the air to sea water CO2 flux as a function of air to sea contact area, wind speed, and pCO2 difference between the seawater and air. These values must be measured in controlled environments to understand their variation with parameters such as the air-sea pCO2 difference. These experiments serve as a stepping stone towards characterizing field pilots in coastal environments, in which many of the dynamics of carbon uptake that can be identified in contained tanks will be lost to ocean mixing, advection, and natural background variation.

**R1C2:** From a scientific perspective, the manuscript is disappointing. The title is misleading and the introduction reads like an appraisal of electrochemical OAE. Potential advantages in scaling considerations are emphasized (line 68ff),  although these are irrelevant for the NaOH experiments performed in this study. Potential disadvantages, such as the treatment of large amounts of acid generated, are not mentioned. This is not what I would expect from a 'comprehensive' assessment, which should, in my view, also include processes in the chambers, effects related to pre-treatment of water, and consideration of the fate of the acid(s) produced.

**R1R2:** See responses R1Ra and R1Rc above.

**R1C3:** The manuscript reads like a progress report of a collection of experiments. The experimental setting in the form of small and large indoor tanks with fluctuating atmospheric CO2, gentle mixing and in some cases bubbling, makes it difficult to accurately and quantitatively compare results.

**R1R3:** Figures 4, 5, and 6 of the original manuscript were meant to provide comparison between experiments conducted at tank scales (Fig. 4), aquaria (Fig. 5), and between tank and aquaria (Fig. 6). As noted in response R1R1d, the choice of experimental setting was chosen for practical sensing and

sampling reasons. Variations between atmospheric/ wind conditions and air bubbling were iterated on throughout the experiments to increase the rate of CO2 equilibration, not to evaluate the kinetics of such changes, but to allow a greater throughput of experiments (averaging 6.5 weeks). All cases shared the same experimental goal, to quantify the storage of atmospheric CO2 resulting from the addition of alkalinity as seawater approached equilibrium with the atmosphere. To clarify this point, we added text at line 232 of the original manuscript, noting: "Initial tank experiments were conducted with a still surface condition, i.e., with no visible water movement across the surface of each tank. As experiments progressed, forced air movement was added across the surface of each tank using a stationary fan with a wind speed of ~5 kph. This was done to control for potential variations in the laboratory HVAC system and to potentially reduce the time to equilibration for the experiments by increasing the rate of air-sea CO2 equilibration. In later experiments, air was bubbled into the bottom of each tank at a rate of ~ 30 L min-1 with an estimated surface area of ~ 0.3 m2, with a goal of further increasing the rate of equilibration to allow for more rapid throughput of experiments."

To further summarize experiments, we have made some changes to Figure 6 of the original manuscript, now Figure 5, in which we compare the results of tank and aquaria experiments with the magnitude of the alkalinity addition as an x-axis for each subplot instead of the initial target pH, and in which we clarified the delta pH in panel c to show the difference in the final pH between the experiment and control data.

**R1C4:** Overall, the experiments show a CO2 removal from the atmosphere that corresponds to an efficiency of about 0.75 moles CO2 taken up per mole of alkalinity added. 'R'unaway precipitation of MgCO3 and CaCO3 was observed, but no quantitative results are shown. How is runaway defined? What stops it?

**R1R4:** 'Runaway precipitation' is a relatively recent term in the context of OAE, referring to a condition where significantly more alkalinity is removed by mineral precipitation than was initially added to seawater (see for example, Moras et al. 2022). Characterization of this precipitation is a hot topic in this field because it is currently poorly defined. We have added text and references to section 3.3 to clarify this description. Specifically at lines [516-519] of the original manuscript, we revised the following text: "However, in cases where enough NaOH was added to raise the bulk seawater $pH_T$ to greater than 10.0 (i.e., in one large tank test with a target $pH_T$ of 10.3, and in 4 aquaria experiments ranging from $pH_T$ 10.0-10.3), runaway precipitation of $Mg(OH)_2$ and $CaCO_3$ was observed. This was characterized by a sharp reduction in both TA and DIC and an increase in turbidity, and a continued depletion of DIC and slow removal of TA as atmospheric $CO_2$ from gas exchange was converted to additional $CaCO_3$. Runaway precipitation has been described as a condition in which more alkalinity is removed from seawater by mineral precipitation than was initially added until a new steady state is achieved (Moras et al., 2022; Hartmann et al., 2023; Suitner et al., 2023). This can significantly reduce the efficiency of OAE, and has implications for biological productivity, as increased turbidity may impact photosynthesis or predator-prey interactions."

Additionally, we have added a discussion of comparable studies relating to runaway precipitation. Beginning at line 554 of the original manuscript, we adapt our text to: "In summary, the presence and duration of brucite precipitation upon addition of 0.5 M aqueous NaOH depends on the ratio of the NaOH addition rate to the local dilution rate in the receiving waters. Future research using flow through tanks could help identify thresholds below which brucite precipitation can be avoided or limited, and the

immediate formation of $Mg(OH)_2$ may be reversible, as also noted by Suitner et al. (2023) and Cyronak et al. (2023). At the given initial seawater conditions, the threshold for aragonite precipitation began at an $\Omega_{arag}$ of 30, corresponding to $pH_T$ > 10.0, and continued as $\Omega_{arag}$ decreased to ~5.2. This threshold corresponded to an increase in TA of >2270 µmol/kg. The potential for runaway aragonite precipitation may be reduced by active mixing at the point of NaOH introduction, maintaining a mixing volume below bulk seawater $pH_T$ of 10.0, and allowing for appropriate dilution in flow-through conditions, particularly on timescales of hours after alkalinity addition.

Characterization of runaway precipitation thresholds at varying temperatures, salinities, and suspended particulate conditions will allow for OAE implementation designs that best avoid this potential risk to OAE efficiency and ecosystem perturbation. We note that these results are only valid for open experiments using an aqueous hydroxide feedstock for alkalinity, and may not be comparable to bench-scale experiments such as closed bottle incubations, where increased surface area, edge effects, and sustained conditions of high $\Omega_{arag}$ may result in precipitation at different thresholds. We also note that we do not assume zero aragonite precipitation at conditions below the stated thresholds, but that potential precipitation is not readily detectable with our experimental setup. For example, heterogeneous $CaCO_3$ precipitation events, such as may occur on suspended sediments in the water column, have been suggested through characteristic changes in seawater TA/DIC ratios in cases of riverine inputs and bottom sediment resuspension (Bustos-Serrano et al., 2009; Wurgaft et al., 2016; 2021). Suspended sediments in the context of OAE project sites could influence OAE efficiency and the potential for runaway precipitation and should be included in future studies (Bach, 2023). The thresholds determined in this study are significantly higher than those of some mineral-based OAE studies, including precipitation after an increase in TA of ~500 µmol/kg using CaO and $Ca(OH)_2$ mineral additions (Moras et al., 2022). Hartmann et al. (2022) noted precipitation resulting from alkalinity additions of >600 µmol/kg $Mg(OH)_2$, and found that aqueous alkaline solutions avoided carbonate precipitation better than reactive alkaline particle additions to seawater. Suitner et al. (2023) suggested that alkalinity additions > 2000 µmol/kg could be achievable given sufficient dilution to avoid runaway precipitation. Together, these studies highlight the need to expand research into runaway precipitation to characterize potential inefficiencies in OAE, particularly in in-situ experiments to establish relationships applicable to ocean environments."

**R1C5:** The introduction states that pH remains 'slightly higher' (line 137) after equilibration with atmospheric CO2. This is also found in the experimental results. It would be valuable to learn how large the increase in pH, and thus the residual mitigation of acidification, is, e.g. in comparison with historical acidification of ocean surface waters. The authors mention that future work should explore this in more detail, and I wonder why this is not already done with the results described here.

**R1R5:** See response R1R1b and R1R3 above. We have adjusted the original Figure 6 to better display the slight increase in pH relative to control experiments. This study features a valuable set of control experiments conducted simultaneous to the experimental additions of alkalinity at large scales. This allows for some comparison between the final pH conditions with and without alkalinity additions, but we note that these controls have some limitations in potential variations over weeks of CO2 equilibration in an open laboratory setting, and that for practical purposes, some experiments were terminated before full equilibration was reached, requiring estimation and some uncertainty in final pH conditions. Improving these estimates would be a useful subject of future work, as would comparison to

historical acidification of ocean surface waters, which we consider to be out of scope of this manuscript.

**R1C6:** The quality of the manuscript is relatively poor. Figures are difficult to read, and some potentially interesting results shown are not discussed in the text, e.g. the > 100% equilibration in figure 5h.

**R1R6:** Specific suggestions on figure improvements would be appreciated, but in response to the general comment of poor figure quality, we have increased the font size in the original Figures 4, 5, and 6, and rearranged Figure 7 for better viewing. In reference to the >100% equilibration in Figure 5h, we originally noted in the text (lines 473-475 of the original manuscript) that there are significant uncertainties in equilibrium estimates due to limitations in seawater carbonate chemistry measurements in small aquaria. We have added text to call out this specific anomaly, with further discussion in supplementary materials. Specifically, we have revised the following text at lines 473-474 of the original manuscript: "We note that there are significant uncertainties in these equilibrium estimates leading to estimates of >100% equilibration. These estimates would be better constrained with more continuous carbonate chemistry measurements, particularly seawater and atmosphere $pCO_2$ throughout the experiments that would allow for more direct calculation of air-sea $CO_2$ flux and equilibration, and finer control of bubbling and diffusion rates are necessary to define the timeline for equilibration within the aquaria."

**R1C7:** I cannot recommend publication of the manuscript in its present form. It would require a new introduction that relates to the experiments and does not promote SEA MATE procedure, that is not an essential element of anything shown in the manuscript. The results obtained are, in principle, scientifically valuable and should be published. They provide relevant and important information about the stability of added alkalinity and show useful attempts to avoid secondary precipitation.

**R1R7:** We appreciate the reviewer's suggestions for the publication of our results and improvements to the introduction to be more consistent with the overall scope of the manuscript. As mentioned above (responses R1R1a and R1R1c), we have reworked the introduction to focus more narrowly on the specific context of these experiments. We have also clarified the purpose of these experiments, which is to lay the groundwork for electrochemical OAE field pilots by providing a relevant but controlled environment in which to gather data that will not be attainable in field trials due to the complexity of the open environment.

**R1C8**: The added value of the large (6000 L) tanks relative to the smaller (16 L) ones is unclear. Are results different? Was there a hypothesis that results should be different? This should be discussed in more detail.

**R1R8:** See also response R1R1d, where we state: "The large tanks allow us to routinely subsample the seawater for the needed carbonate chemistry measurements at the required precision to evaluate this method. They also allow for deployment of larger oceanographic sensors than can be submerged in smaller experiments. The smaller aquaria are primarily employed to allow us to generate multiple replicate measurements of the $CO_2$ removed to alkalinity ratio, that is the OAE efficiency." We also note in lines 206-208 of the original manuscript that the larger tanks were chosen to limit interactions with walls, and add that "The large tank volumes were chosen to limit interactions with walls while increasing the air-seawater boundary, and to lose a smaller fraction of their volume to evaporation." In both respects, the tanks are a closer representation of the real ocean than bench scale experiments.

The purpose of the smaller experiments is described at the beginning of the methods section in lines 211-215 of the original manuscript and repeated in the aquaria results section in lines 433-437 over the original manuscript — the long equilibration times (~6.5 weeks) in the large tanks limited our capacity to vary experimental conditions and measure many replicates under different experimental conditions. Smaller aquaria were used to allow for additional test cases, albeit with reduced sensing capacity due to the size and number of instruments available (lines 258-259 of the original manuscript). Different results are not expected from the large tanks as compared to the smaller aquaria. We have expanded on this after line 215 of the original text, adding: "While it is expected that equilibration occurs more rapidly in the small aquaria than in the large tanks, the results from these cases should be similar as CO2 equilibrates across the air-sea boundary. However, we note that some variation is expected due to limited sensing and sampling options in the smaller aquaria and the greater potential for biological growth in the large tanks over longer timescales."

**R1C9:** Given the urgency of the climate problem and the demand on sound scientific information about OAE (and other CDR approaches), the results of this set of experiments should to be published. There are several places where the authors seem to indicate that the experimental setting was not ideal. Such an open sharing of information is much appreciated. Additional value could be gained by a brief discussion of what the authors would recommend for a possible next round of experiments.

**R1R9**: We appreciate the referee's assessment that this work should be published and also appreciate their comments and suggestions on the open sharing of information and description of future research needs. We direct the referee to the summary of the original manuscript where we suggest ongoing and future work in regards to constraints on mixing, sensing improvements, characterization of precipitation, biological control, acidification considerations, and the need for mesocosm and field experiments (lines 569-573, 581-589, 599-605, 609-613, 614-618, and 625-627, of the original manuscript respectively).

I recommend major revision.

individual points:

**R1C10**: l.45 I don't think that IPCC 2022 gives annual removal rates in Gt CO2/yr, but, as far as I remember, "only" a range of required total removal until the end of century. This could be converted in some average rate, but as written here seems to pretend an accuracy and precision that cannot justified by referring to the IPCC report.

**R1R10:** For clarity, we have edited the text to recognize that the IPCC report and following studies have suggested a range of potential carbon removals required. Specifically, in the revised manuscript, lines 44-48 now read: "The Sixth Assessment Report of the Intergovernmental Panel on Climate Change reported that in addition to a drastic decrease in CO2 emissions, active removal of 5-15 Gt of atmospheric CO2 per year by 2100 is necessary to constrain average global warming to less than 1.5 - 2 °C (noting that the magnitude of carbon removals varies by climate scenario: IPCC, 2022; Rogelj, 2018)."

**R1C11**: l.97 ff the text is difficult to understand. Why does the concentration of H+ not change? Would this mean one could extract an infinite amount of H+ from seawater? There must be some adjustments that may be small for small extractions, but the language here suggests an infinite H+ reservoir.

**R1R11:** This text references the chemistry within a specific electrochemical process that produces NaOH from water. This section has been removed from the text because it is outside the scope of the revised manuscript, as discussed above in response R1R1c.

**R1C12:** l. 117 this equation is not balanced and presumably lacks water on the left hand side?

**R1R12:** As part of the simplification for the introduction, we have removed this passage.
* * *
**RC2:** 'Comment on egusphere-2024-108', Anonymous Referee #2, 04 Mar 2024

Review of Ringham et al., 2024:

Ringham et al. investigate atmospheric CO2 influx and associated carbonate chemistry changes following an alkalinity enhancement perturbation with NaOH. They also investigate secondary precipitation responses to the perturbation. The methods appear robust. However, it remained unclear until the end what critical research question this study aimed to address. Please find below my main comments and some additional ones. A possible way forward is mentioned after the main comments.

Main comments:

**R2C1:** It is unclear what critical knowledge gap is addressed here. Air-sea CO2 fluxes are a relevant topic to study but the problem with it is primarily how complicated ocean physics (dispersion/dilution/subduction) drive equilibration curves (not primarily that stirring accelerates influx). This critical question concerning ocean physics cannot be addressed in this laboratory setting which lacks realistic representation of water mass movement, dilution, and subduction. As such, this study can only show that CO2 equilibration with the atmosphere takes more than days (already known) and that bubbling/stirring accelerates influx (already known). There appears to be no potential for novel insights here.

**R2R1:** We disagree with this comment on critical knowledge gaps. Currently, our understanding of the impacts of NaOH-based ocean alkalinity enhancement on seawater carbonate chemistry is mostly rooted in benchtop-scale experiments and modeling exercises. As marine carbon removal efforts move forward, we expect to see small pilot trials releasing alkalinity into open ocean environments at scales that will be very challenging to monitor with existing carbonate chemistry technology. The experiments presented in this manuscript show a scale-up as a stepping stone from bench-scale work towards field trials, to provide context for pilot experiments when carbonate chemistry signals related to OAE dynamics are lost in natural background variability. These experiments, paired with modeling, will form the necessary basis for measurement, reporting, and verification (MRV) of carbon removal from electrochemical OAE projects. Hypothesis testing at the scale of controlled laboratory tanks is particularly important in allowing for comparison between experiment and control cases, allowing for direct quantification of additionality and uncertainty in the absence of factors including biological complications, NaOH generation, and natural background variability, which are all targets of future work in advancing towards field trials.

**R2C2:** The manuscript is much too long. The amount of information provided does not justify the amount of text. It is most obvious in the introduction where many other aspects than studied here are introduced. But also the remaining bits can be shortened, since the physical/chemical processes described here are already largely understood.

**R2R2:** Agreed. We have removed extraneous text from the original manuscript that described details of the electrochemistry that could be used to produce aqueous alkalinity. Specifically, we have deleted lines 72-90 and Fig. 1 from the original manuscript that described details of the electrochemical process to generate alkalinity. These changes tightened up the manuscript and also made it more consistent with the scope of the manuscript.

**R2C3:** The text is currently written more like a report or research thesis than a paper. It lacks a clear definition of knowledge gaps and how the study can make progress on them. At the end it lacks a reflection of what was learnt und where we stand now (after the study). Adding that would help to understand the knowledge gain.

**R2R3:** Regarding knowledge gaps, we direct the reviewer to lines 180-189 in the introduction of the original manuscript where we describe key questions in OAE addressed in this work. While carbon uptake by seawater is generally understood, there is limited experimental literature specific to NaOH-based OAE. Experiments of this type provide necessary context for upcoming pilot experiments in which we expect that seawater carbonate chemistry signals in response to small alkalinity additions will be challenging to extract from the natural background variation in coastal environments.

Regarding a reflection of what was learned: The original manuscript provides a summary highlighting the needs for future work in this field, including critical needs in understanding safe and efficient thresholds for OAE deployments. Specifically the summary of the original manuscript suggests ongoing and future work in regards to constraints on mixing, sensing improvements, characterization of precipitation, biological control, acidification considerations, and the need for mesocosm and field experiments (lines 569-573, 581-589, 599-605, 609-613, 614-618, and 625-627, of the original manuscript respectively). In addition, we have added the following text at line 560 of the revised manuscript, addressing runaway precipitation: "Characterization of runaway precipitation thresholds at varying temperatures, salinities, and suspended particulate conditions will allow for OAE implementation designs that best avoid this potential risk to OAE efficiency and ecosystem perturbation. We note that these results are only valid for open experiments using an aqueous hydroxide feedstock for alkalinity, and may not be comparable to bench-scale experiments such as closed bottle incubations, where increased surface area, edge effects, and sustained conditions of high $\Omega_{arag}$ may result in precipitation at different thresholds. We also note that we do not assume zero aragonite precipitation at conditions below the stated thresholds, but that potential precipitation is not readily detectable with our experimental setup. For example, heterogeneous $CaCO_3$ precipitation events, such as may occur on suspended sediments in the water column, have been suggested through characteristic changes in seawater TA/DIC ratios in cases of riverine inputs and bottom sediment resuspension (Bustos-Serrano et al., 2009; Wurgaft et al., 2016; 2021). Suspended sediments in the context of OAE project sites could influence OAE efficiency and the potential for runaway precipitation and should be included in future studies (Bach, 2023). The thresholds determined in this study are significantly higher than those of some mineral-based OAE studies, including precipitation after an increase in TA of ~500 µmol/kg using CaO and $Ca(OH)_2$ mineral additions (Moras et al., 2022). Hartmann et al. (2022) noted precipitation resulting from alkalinity additions of >600 µmol/kg $Mg(OH)_2$, and found that aqueous alkaline solutions avoided carbonate precipitation better than reactive alkaline particle additions to seawater. Suitner et al. (2023) suggested

that alkalinity additions > 2000 µmol/kg could be achievable given sufficient dilution to avoid runaway precipitation. Together, these studies highlight the need to expand research into runaway precipitation to characterize potential inefficiencies in OAE, particularly in in-situ experiments to establish relationships applicable to ocean environments."

**R2C4**: The results/discussion + summary sections contained just one reference. I double-checked and could really only find one (which was a citation of some methodological detail, not central to the study). Therefore, there was no effort to align the research with what was already known, which strengthened the impression of limited knowledge gain.

**R2R4:** The citation referenced in this comment was line 490-491 of the original manuscript, reading: "In general, the large tanks and aquaria showed reasonable agreement in achieving values for CAR within the expected range of 0.7-0.9 (He and Tyka, 2023)", following from results presented in text, tables, and figures in this study. This is in fact central to this study, addressing the first key question presented in line 180 of our introduction, "(1) How much additional atmospheric $CO_2$ is stored in seawater as DIC in response to a given alkalinity perturbation?" As discussed in R2R1, R2R3, and R2R26, evaluation of carbon storage resulting from alkalinity additions at increasing scales from benchtop experiments through mesocosms form the basis of knowledge required to progress towards field trials, in which we do not expect to be able to monitor for this CAR ratio in real ocean conditions. In the absence of direct measurements of carbon storage from OAE, it is likely that experimental results of this type paired with simulated dilution of alkalinity into the real surface ocean will form the basis of MRV for this mCDR approach. We have added text to the summary section at line 566 of the original manuscript, "These results are in general agreement with ratios noted in Burt et al. (2021), He and Tyka (2023), and Wang et al., (2023)."

To further align this study with recent work, we note the addition of text on runaway $CaCO_3$ precipitation added at 560 (R1R3). In the summary section, we now direct readers to Zhang and Bryne, 1996, and Liu et al., 2006, in line 589 of the original manuscript on the use of thymol blue for pH measurements, and to the Best Practices Guide to Research in OAE in line 627.

**R2C5:** The title does not at all reflect what was done in the study.

**R2R5:** Agreed. The scope of the manuscript is limited to an assessment of the response of the ocean atmosphere system to the enhancement of alkalinity in the form of aqueous sodium hydroxide. Aqueous sodium hydroxide is the form of alkalinity generated by electrochemical ocean alkalinity approaches, hence the original title. However we understand in retrospect how this title could lead the reader to assume we are assessing the effectiveness of the electrochemical process that generates the alkalinity. As a result we have revised the title to "An assessment of ocean alkalinity enhancement using aqueous hydroxides: kinetics, efficiency, and precipitation thresholds."

**R2C6**: Overall, the study needs a major overhaul. It can probably be shortened by ~50%, for example by deleting all text addressing issues outside the context of the dataset. Goals and especially key knowledge gaps to be addressed here need to be clearly defined. The achievements need to be critically evaluated to justify if the study really made progress on defined knowledge gaps. The discussion should contrast findings to available research. This will also help to evaluate the study's novelty, which so far is entirely unclear.

**R2R6:** As noted in responses include R2R4, text has been added to the revised manuscript to more directly relate findings to available research, but we note that there are limited studies available for comparison. To the best of our knowledge, this is the first study scaling NaOH-based alkalinity additions to natural seawater between control and experiment tanks at high resolution of carbonate chemistry, filling an important gap between bench-scale studies and pilot trials. The goals of this study are defined in lines 180-189 of the original manuscript, calling for (in brief): 1) quantification of atmospheric $CO_2$ storage in seawater following an NaOH-perturbation, 2) timescale of $CO_2$ removal and storage, 3) assessment of $Mg(OH)_2$ precipitation, and 4) thresholds for $CaCO_3$ precipitation, each of which are addressed throughout this study. Future work is needed to better quantify each of these questions, which are necessary for understanding OAE in pilot trials in the real ocean where quantification of each factor will be convoluted by natural background variation and the long timescale of air-sea $CO_2$ equilibration required.

Additional comments (some of which overlap with the major comments):

**R2C7:** Introduction: The introduction can be condensed substantially. I would think at least 50% (probably more) can easily be removed. For example all the descriptions of the different OAE methods and also the basic chemistry around alkalinity and eta. All this can be eliminated by appropriate referencing.

**R2R7:** We agree that the introduction can be condensed and have removed text specific to electrochemical OAE methods. However, because this is a fairly new and interdisciplinary field, we believe that some context in OAE methods and seawater carbonate chemistry, particularly total alkalinity, is warranted to set the context for this study as a stepping stone between bench and pilot scale experiments.

**R2C8**: Line 45: Scenarios suggest a range of numbers (depending on mitigation success), not just 5.5Gt/year.

**R2R8:** We thank the reviewer for this flag. For clarity, we have edited the text to recognize that the IPCC report and following studies have suggested a range of potential carbon removals required. Specifically, in the revised manuscript, lines 44-46 now read: "The Sixth Assessment Report of the Intergovernmental Panel on Climate Change reported that in addition to a drastic decrease in $CO_2$ emissions, active removal of 5-15 Gt of atmospheric $CO_2$ per year by 2100 is necessary to constrain average global warming to less than 1.5 - 2 °C (noting that the magnitude of carbon removals varies by climate scenario: IPCC, 2022; Rogelj, 2018)."

**R2C9:** Line 47 and elsewhere: Perhaps reduce words that enforce a statement (here "enormous"). It is all relative.

**R2R9:** We accept the recommendation to replace non-quantitative descriptions and have removed this term—however, we stand by the sentiment that the scale of projected necessary carbon removal remains an enormous challenge.

**R2C10:** Line 54: Scalability is not yet clarified as it depends on economic/engineering etc. not primarily on how many moles modelers added in their models.

**R2R10:** We do not argue that scaling depends only on modeled scenarios, but specifically call out that development requires consideration of many economic and engineering factors, which include alkalinity source, delivery, and location (lines 54-57 of the original manuscript).

**R2C11**: Lines 130-140: There is some text which sounds a bit like a marketing pitch. Can probably deleted as the relevance question under investigation appears to be the air-sea CO2 influx.

**R2R11:** This is a fair comment. We note that this is a major research topic in this field— OAE has been noted for potential co-benefits in local acidification mitigation, but little research is available to evaluate this possibility. Understanding the potential negative and positive impacts on marine communities is a necessary aspect of evaluating the scaling and siting opportunities for OAE. We have removed lines 130-140 from the original manuscript to reflect the scope of this study.

**R2C12**: Line 167: Albright et al studied ocean acidification. If NaOH experiments out of other contexts are repurposed here then it would also be necessary to consider all the NaOH-based carbonate chemistry manipulation work from the 90ies and early 2000s where CO2 limitation in various species (phytoplankton/corals) was studied.

**R2R12**: We direct the reader to Albright et al., 2016 as one of the most comparable releases of NaOH into an open environment. While this work was not done with the intention of investigating OAE impacts, it is a useful reference point as we work towards OAE pilots. We have modified the text to clarify this point. Lines 165-168 of the original manuscript now read: "While some work has investigated various aspects of NaOH-based ocean alkalinity enhancement in microcosms (Ferderer et al., 2022; Hartmann et al., 2023) and mesocosms (Groen et al., 2023), and other work has studied the release of NaOH over natural coral reefs as a method of local ocean acidification mitigation (Albright et al., 2016), a systematic characterization of the efficiency and kinetics of OAE as a function of key process parameters has not yet been performed."

**R2C13:** Line 184: Why spatial scale in the tank?

**R2R13:** It is very challenging to extrapolate spatial and temporal scales of mineral precipitation and dissolution from tanks to field work. However, we expect that it will be even more difficult to identify mineral precipitation and dissolution in dynamic field trials at small pilot experiment scales. Understanding this behavior is important to designing pilots that effectively deploy alkalinity to capture atmospheric carbon with minimized local impacts on biology, such as may occur under increased turbidity from Mg(OH)2 precipitation. This is expanded on in later sections, including the results, where we note that some Mg(OH)2 always precipitates upon introduction of NaOH to seawater, but that it also redissolves rapidly (within seconds and in the immediate proximity of the NaOH release) in all but still water. In line 184 of the original manuscript, we have simplified this question to: "What are the conditions for Mg(OH)2 precipitation upon addition of NaOH to seawater?"

**R2C14:** Line 205: How can the tank be 0.24 m in diameter and have a surface area of 4.6 m2? This part of the description needs a clean-up, perhaps a table.

**R2R14:** We thank the reviewer for catching this typo! The diameter was 2.4m.

**R2C15:** Line 215: has been said before.

**R2R15:** We suspect that the reviewer is referring to line 208-209, where we describe the limitations of the large tank tests. Line 215 is specific to aquaria limitations.

**R2C16**: Line 218: Did the bleach affect alkalinity?

**R2R16**: This is an excellent question—the bleach was not found to affect alkalinity within the precision of bottle sampling. We added a note to the text to clarify. Specifically, line 223 of the original manuscript now reads "Measurements of total alkalinity showed no significant differences in the bulk seawater TA before and after the bleaching process.".

**R2C17**: Line 231: "still" as "calm"?

**R2R17:** Correct—we use the term 'still' throughout the manuscript to specify that there was no visible water mass movement at the surface of the tank.

**R2C18**: Line 335: Lueker.

**R2R18:** We thank the reviewer for catching this typo, which was correctly noted in the references.

**R2C19**: Line 359: To what extent do the salinity increases affect the saturation state and to which extent does this complicate the representativeness of the results?

**R2R19**: This is a good question. Due to evaporation, the salinity, TA, and DIC of each tank increased over time in each experiment. The order of data processing is described in the text in Section 2.4, and in brief, includes normalization of bottle sample DIC and TA to salinity, followed by calculation of pCO2, pH, and aragonite saturation state using CO2sys. We present saturation state values calculated using the salinity-normalized TA and DIC (normalized to the initial salinity measured at the beginning of each experiment) to represent expected values in the absence of evaporation.

**R2C20:** Figure 3: k is meant to show "CAR" but then the axis says something else?

**R2R20**: The axis of panel k (delta_nDIC/delta_nTA) is simplified for readability, and is described more fully in the caption as CAR, i.e. (nDICexp – nDICcontrol)  / ΔTANaOH addition.

**R2C21**: Line 410: This conclusion assumes that everything non-alkalinity related is identical between the control and treatment tank. But this may not be valid in case chemistry/biology evolve differently between tanks (either randomly or due to alkalinity).

**R2R21:** This is a good point. It is worth noting, however, as stated on lines 351 - 353 of the original manuscript "The experiments were subsequently refined to allow for several days of cross-pumping between tanks to homogenize the control and experiment seawater before NaOH was added at the start of an experiment." Nevertheless,we did oversimplify this statement to say that the control is needed to capture natural dynamics, but did not delve into potential differences between the tanks. We have added a line to expand this point. Specifically, line 354 of the original manuscript now reads "While the initial seawater conditions were similar between the control and experiment tanks, we note that these cases are not entirely comparable after the termination of cross-pumping between tanks and the subsequent addition of alkalinity. While tanks were initially bleached, eventually some biological growth

was noted in each tank with potential differences in spatial and temporal distribution as well as species and community differences. Herein, we assume that differences between the control and experiment cases are due to the addition of alkalinity alone, but we note that characterization of other potential confounding factors is a subject for future work."

**R2C22:** Figure 4: I may have overlooked it but what caused the TA increase in 8.5 FAM around day 30?

**R2R22**: This is a good question. The TA increase was noted in both the control and experimental tanks, and is not yet understood. There is a description of this event in the supplementary materials (Section S4), and we have added a note to the main text after line 400 of the original manuscript, which now reads: "An anomalous event was noted in both the experiment and control cases for the target pHT 8.5 experiment with forced air movement across the surface of the tank, wherein an increase in TA and DIC was noted around day 30 of the experiment. The cause of this event is unclear but could include biological changes in both tanks, the introduction of alkalinity from environmental contaminants, or the anomalous delayed release of alkalinity from suspended solids. This event was not observed in any other case, and highlights the importance of using controls to understand complex interactions in these experiments. A time-series including this event is available in the Supplementary Materials."

**R2C23:** Figure 4: is this TA or nTA? Asking because it is hard to see if the values are stable or not (due to evaporation).

**R2R23:** The DIC and TA are normalized throughout this plot—we have updated the figure labels and caption to reflect this, and have increased the font size throughout the figure.

**R2C24**: Line 557: This appears to be a simplification. Conditions in the other tanks were not stable (they allowed CO2 influx) and there may have been precipitation if they water had been under the high Omega conditions for longer. I get that authors say "under these conditions" but I would nevertheless recommend some more clarity here as what sticks is that "pH 10 is the threshold".

**R2R24:** We agree that conditions throughout the other experiments were not stable and that precipitation could have occurred if they had retained higher omega conditions for longer, but they did not retain the high pH and omega signals because of the CO2 influx as the tanks were open to the atmosphere. Closed experiments would not have served the purpose of this study. Nevertheless, we have added some text for clarity, and additionally note that we are not stating that zero precipitation occurs below a threshold of pH 10 for these experiments, but that no measurable precipitation occurs. Specifically, lines 554 - 560 of the original manuscript have been adapted to the following: "In summary, the presence and duration of brucite precipitation upon addition of 0.5 M aqueous NaOH depends on the ratio of the NaOH addition rate to the local dilution rate in the receiving waters. Future research using flow through tanks could help identify thresholds below which brucite precipitation can be avoided or limited, and the immediate formation of $Mg(OH)_2$ may be reversible, as also noted by Suitner et al. (2023) and Cyronak et al. (2023). At the given initial seawater conditions, the threshold for aragonite precipitation began at an $\Omega_{arag}$ of 30, corresponding to $pH_T > 10.0$, and continued as $\Omega_{arag}$ decreased to ~5.2. This threshold corresponded to an increase in TA of >2270 µmol/kg. The potential for runaway aragonite precipitation may be reduced by active mixing at the point of NaOH introduction, maintaining a mixing volume below bulk seawater $pH_T$ of 10.0, and allowing for appropriate dilution in flow-through conditions, particularly on timescales of hours after alkalinity addition.

Characterization of runaway precipitation thresholds at varying temperatures, salinities, and suspended particulate conditions will allow for OAE implementation designs that best avoid this potential risk to OAE efficiency and ecosystem perturbation. We note that these results are only valid for open experiments using an aqueous hydroxide feedstock for alkalinity, and may not be comparable to bench-scale experiments such as closed bottle incubations, where increased surface area, edge effects, and sustained conditions of high $\Omega_{arag}$ may result in precipitation at different thresholds. We also note that we do not assume zero aragonite precipitation at conditions below the stated thresholds, but that potential precipitation is not readily detectable with our experimental setup. For example, heterogeneous $CaCO_3$ precipitation events, such as may occur on suspended sediments in the water column, have been suggested through characteristic changes in seawater TA/DIC ratios in cases of riverine inputs and bottom sediment resuspension (Bustos-Serrano et al., 2009; Wurgaft et al., 2016; 2021). Suspended sediments in the context of OAE project sites could influence OAE efficiency and the potential for runaway precipitation and should be included in future studies (Bach, 2023). The thresholds determined in this study are significantly higher than those of some mineral-based OAE studies, including precipitation after an increase in TA of ~500 µmol/kg using CaO and Ca(OH)$_2$ mineral additions (Moras et al., 2022). Hartmann et al. (2022) noted precipitation resulting from alkalinity additions of >600 µmol/kg Mg(OH)$_2$, and found that aqueous alkaline solutions avoided carbonate precipitation better than reactive alkaline particle additions to seawater. Suitner et al. (2023) suggested that alkalinity additions > 2000 µmol/kg could be achievable given sufficient dilution to avoid runaway precipitation. Together, these studies highlight the need to expand research into runaway precipitation to characterize potential inefficiencies in OAE, particularly in in-situ experiments to establish relationships applicable to ocean environments."

**R2C25:** The "results and discussion" section reads like only results (evident by no comparison with other studies or any referencing). Only the very last paragraph contains some discussion.

**R2R25:** This is a fair point—we have added some discussion text at the end of sections 3.1 and 3.2 to tie these results to the following summary. See R2R4.

**R2C26**: Line 562: One could argue that the key finding expressed here was already known and that the timescales are not transferrable to the real ocean. I think the key issue with the study is that it either does not explain well enough what its novelty is, or it does not contain novelty relevant for alkalinity enhancement in the ocean.

**R2R26:** We disagree with this comment. Broadly, not much is known about alkalinity enhancement. We are currently seeing the rapid development of a new field with as yet to be determined commercial, climate, and environmental implications. There are only a handful of active and upcoming pilot studies planned for various OAE implementations and locations, and small pilot scales will be inherently challenging to characterize as they add minimal amounts of alkalinity to dynamic coastal environments and typically lack a viable control experiment. These signals will be initially designed for minimal measurable perturbation as we learn more about impacts on marine communities, and it is expected that carbon uptake signals will be lost to natural background variability over the weeks to months of equilibration required for air-sea gas exchange. Because of this, measurement, reporting, and verification of carbon removal from OAE deployments in the open ocean will rely on models, which will in turn be informed by experimentation on factors including OAE efficiency, kinetics, biological impacts, and precipitation thresholds.

The large tank experiments presented in this manuscript provide a stepping stone between bench-scale experiments and in-situ mesocosms or field pilots. Even if these experiments simply confirm stoichiometric and modeled expectations, this is critical information in the design and implementation of OAE deployments. This work is a necessary part of the growing scientific body that will allow for field trials to progress.

**R2C27:** Line 574-589: This paragraph largely adds another discussion point, not a summary item. (same for next paragraph).

**R2R27**: We understand the reviewer's comment, but argue that these passages are less of a discussion of the results presented and more a discussion towards necessary future work, as is appropriate to a summary section.
* * *
Finally, we summarize the changes to our manuscript during this review:

1) Adjustment of the original title to "An assessment of ocean alkalinity enhancement using aqueous hydroxides: kinetics, efficiency, and precipitation thresholds"
2) Simplification of the introduction to remove discussion of SEA MATE project at large, to focus of the scope of this study as an in-between stepping stone from bench to field trials
3) Simplification of methods details from the Results and Discussion section and rearrangement into the Methods section
4) Adjustment of all figures for increased font size
5) Adjustment of original Figure 6 to place data in context of alkalinity addition (x-axis) instead of target pH
6) Additional discussion of mineral precipitation thresholds in context of prior studies in the discussion section
7) Addition of multiple references
8) Correction of minor typos and grammatical improvements

We believe these changes have made for a substantially improved manuscript, and thank the editors and reviewers for their comments.

---

## Author Response (AR2)

To the reviewers and editors:

We thank you for your second review of "An assessment of ocean alkalinity enhancement using aqueous hydroxides: kinetics, efficiency, and precipitation thresholds." The initial round of comments led to major improvements in the manuscript, and the latest review has continued the trend towards a stronger and more efficient description of our work.

Below in our Response to Reviewers, we provide a point-by-point response to reviewer comments and explain the changes that were made to the manuscript, provided in track-changed and clean formats.

On behalf of all the co-authors,
Sincerely,
Dr. Mallory Ringham
Adjunct, Stony Brook University
Lead Oceanographer, Ebb Carbon, Inc.
* * *
**Responses to reviewer comments**
In the responses, "[Line N]" indicates that the location is line number N in the track-changed version of the latest reviewed version of the manuscript.
* * *
**Report #1: Anonymous referee #2**

**Comment 1:** The authors have addressed all comments. Their argument against "unclear novelty" is that tank scale experiments are a necessary transition from benchtop-scale to field studies. I partially agree, it is indeed desirable to show results like the ones presented here to confirm approximate kinetics of gas exchange and secondary precipitation derived from other studies and different scales. I only "partially" agree because I think their argument that not much known is for CO2 exchange in the context of OAE is not a good one. It is unclear why knowledge on air-sea CO2 exchange could not equally well be derived from totally different research fields. Obviously, such research exists and models are parameterized accordingly, which is why the timescales estimated with them are broadly consistent with the ones measured here. Nevertheless, the first argument (there is value of different lab scale experiments) may outweigh the latter, especially when it comes to permitting. The responses to the other comments are also satisfactory, although I was at times surprised by the authors' defensiveness. For example, the authors argued against the comment that there was only one reference in the whole results/discussion/summary sections. I would have thought that there was not much to argue against. But in any case, the authors have enriched these parts and tried to add further context so ultimately considered the recommendation.

**Response 1:** We thank the referee for their comments on the novelty of this topic. We agree—prior research on air-sea CO2 exchange in other fields lends significant confidence to OAE research. The need for demonstration of this process at intermediate scales is driven by an abundance of caution in moving mCDR research from the lab to field trials and is critical to social license and scientific reputation in testing OAE approaches. We appreciate the valuable commentary throughout the discussion section, which we improved through comparison with relevant publications in the previous revision.

**Comment 2:** One remaining (minor) comment is that the summary still sounds more like a discussion. E.g. the repetition of methodological details seems somewhat out of place. Furthermore, the summary contains new discussion points. This is not critical but may be addressed.

**Response 2:** This was a helpful note. In finalizing this manuscript, we shifted several sentences from the Results and Discussion to the Methods, including lines 353-358 (discussion of changes in steps to homogenize control and experiment seawater before beginning an experiment), 452 – 455 (highlighting again the use of smaller aquaria to enable additional experiment cases on shorter timelines, and 474-480 (noting handling of evaporation in aquaria experiments. We have renamed the Summary section to Summary and Future Work, to clarify that discussion points in this section are specifically relevant to improvements and new directions that could be taken from this work.
* * *
**Report #2: Referee #3, Sijia Dong, sijiadong@nju.edu.cn**

**Comment 1:** Ringham et al. present original experimental work on the chemical consequences of ocean alkalinity enhancement. The authors exclude biological responses by pre-treating seawater with bleach and UV, and monitor the water chemistry and solid phase precipitation within the water column. The authors include, in their manuscript, a good incorporation of up-to-date studies, as related researches in ocean alkalinity enhancement have been emerging rapidly. I think this work provides important new data and information for the field, and should definitely be seen by the science and policy-making community soon. Overall, this work provides new experimental data and information to better understand the consequences of ocean alkalinity enhancement, a widely interested topic. I would recommend publishing the manuscript after minor revision.

A small concern I have about the research is:

In Section 3.3, the runaway precipitation section, the authors state that Mg(OH)2 immediately precipitate after NaOH addition, as thick, slurry-like precipitants. Then, within 24 hours of NaOH addition, most Mg(OH)2 signals disappear (from the XRD spectrum of filtered particle samples), and aragonite start to precipitate. TA decrease as the two minerals precipitate, and drop below that of the initial seawater within 24 hours of the NaOH addition (Line 543-544). In Figure 6, there seems to be another really intense and crystalline solid phase in addition to brucite at time zero. The authors do not specify what this phase is. And there seems to be aragonite peak as well (around 2-theta = 28 degrees) at t0. Brucite XRD peaks should be quite narrow and intense too (in Figure 6, at 0 hr and 3 hr, the peaks seem quite broad). But when there are "sharper" peaks, other peaks might be hard to see. In this case, "other peaks" would be brucite and/or aragonite. I would recommend the authors identify what the intense mystery peaks are (for example, it could be bleach that decompose with time?), and see whether removing these "noise signals" affects the interpretation of their results. If the authors still have spare samples, they can analyze the samples again after rinsing. I would also recommend a more quantitative analysis of Mg(OH)2 / aragonite ratio if possible.

**Response 1:** We thank the referee for their comments on this work. In addressing the questions on the XRD data presented in Figure 6, we have revised and updated the figure to include the peak angles characteristic of brucite, calcite, aragonite, vaterite, and NaCl. We have highlighted a few additional peaks showing the progression from brucite-dominated precipitation to aragonite-dominated precipitation over time. We note that the noise in this Figure in the 0 and 3 hour time points is likely due to the difficulty in removing salt from the thick slurry of these samples (see lines 564-566). The suggestion that bleach could be decomposing is an interesting one, but was ruled out through a series of Hach tests of free chlorine and monitoring of pH that suggested the effective decomposition of bleach before the beginning of each experiment. In regards to the 2θ peaks ~27°, we note that there is a single peak could be associated with NaCl at the 0 hour timestamp, and there are two peaks associated with aragonite at time = 3+ hours. We have added two figures to the Supplementary Materials (S7 and S8) from duplicates of this experiment at aquaria scale. While a t = 0 hour sample is only available in one of the two aquaria experiments, we note a similar distribution of peaks that indicate NaCl trending to aragonite over time. While there are no spare samples available for reanalysis, we agree that future studies would benefit from improved slurry collection and cleaning prior to XRD followed by additional analyses of the brucite/ aragonite ratio.

---

## Author Response (AR3)

To the editor:

Thank you for your close review of "An assessment of ocean alkalinity enhancement using aqueous hydroxides: kinetics, efficiency, and precipitation thresholds." We have made minor requested edits including adding subscripts to $CO_2$ and italicizing $pCO_2$ in the introduction. We added a space at line 398, and replaced 'decomposition of biology…bleaching step' with 'decomposition of organic material' in lines 416 and 472.

On behalf of all the co-authors,
Sincerely,
Dr. Mallory Ringham
Adjunct, Stony Brook University
Lead Oceanographer, Ebb Carbon, Inc.